# Interpretable and Explainable Logical Policies via Neurally Guided Symbolic Abstraction

**Quentin Delfosse**[*]
Technical University of Darmstadt
National Research Center for Applied Cybersecurity
quentin.delfosse@tu-darmstadt.de

**Hikaru Shindo**[*]
Technical University of Darmstadt
hikaru.shindo@tu-darmstadt.de

**Devendra Singh Dhami**
Eindhoven University of Technology[†]
Hessian Center for AI (hessian.AI)
d.s.dhami@tue.nl

**Kristian Kersting**
Technical University Darmstadt
Hessian Center for AI (hessian.AI)
German Research Center for AI (DFKI)
kersting@cs.tu-darmstadt.de

## Abstract

The limited priors required by neural networks make them the dominating choice to encode and learn policies using reinforcement learning (RL). However, they are also black-boxes, making it hard to understand the agent's behaviour, especially when working on the image level. Therefore, neuro-symbolic RL aims at creating policies that are interpretable in the first place. Unfortunately, interpretability is not explainability. To achieve both, we introduce Neurally gUided Differentiable loGic policiEs (NUDGE). NUDGE exploits trained neural network-based agents to guide the search of candidate-weighted logic rules, then uses differentiable logic to train the logic agents. Our experimental evaluation demonstrates that NUDGE agents can induce interpretable and explainable policies while outperforming purely neural ones and showing good flexibility to environments of different initial states and problem sizes.

## 1 Introduction

Deep reinforcement learning (RL) agents use neural networks to take decisions from the unstructured input state space without manual engineering [Mnih et al., 2015]. However, these black-box policies lack *interpretability* [Rudin, 2019], *i.e.* the capacity to articulate the thinking behind the action selection. They are also not robust to environmental changes [Pinto et al., 2017, Wulfmeier et al., 2017]. Although performing object detection and policy optimization independently can get over these issues Devin et al. [2018], doing so comes at the cost of the aforementioned issues when employing neural networks to encode the policy.

As logic constitutes a unified symbolic language that humans use to compose the reasoning behind their behavior, logic-based policies can tackle the interpretability problems for RL. Recently proposed Neural Logic RL (NLRL) agents [Jiang and Luo, 2019] construct logic-based policies using differentiable rule learners called $\partial ILP$ [Evans and Grefenstette, 2018], which can then be integrated with gradient-based optimization methods for RL. It represents the policy as a set of weighted rules, and performs policy gradients-based learning to solve RL tasks which require relational reasoning. It successfully produces interpretable rules, which describe each action in terms of its preconditions and outcome. However, the number of potential rules grows exponentially with the number of considered actions, entities, and their relations. NLRL is a memory-intensive approach, *i.e.* it generates a set of potential simple rules based on rule templates and can only be evaluated on simple abstract

---

[*]Equal contribution.
[†]DSD contributed while being with hessian.AI and TU Darmstadt before joining TU\e.

37th Conference on Neural Information Processing Systems (NeurIPS 2023).

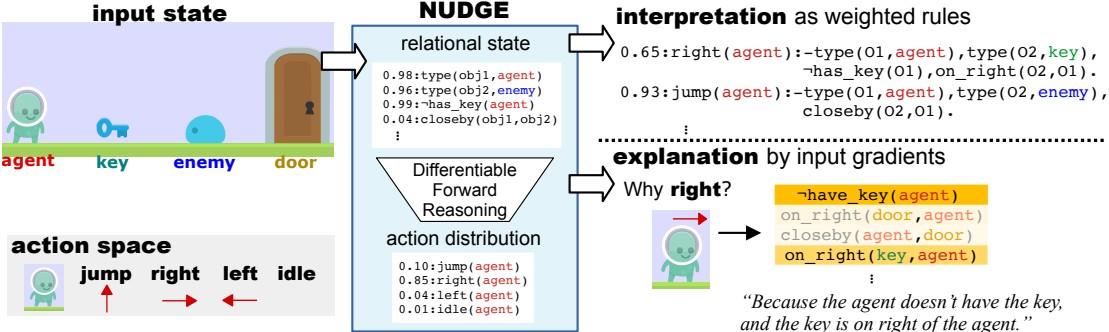

Figure 1: **Overview of NUDGE.** Given a state (depicted in the image), NUDGE computes the action distribution using relational state representation and differentiable forward reasoning. NUDGE provides *interpretable* and *explainable* policies, *i.e.* derives policies as sets of interepretable weighted rules, and can produce explanations using gradient-based attribution methods.

environments, created for the occasion. This approach can generate many newly invented predicates without their specification of meaning [Evans and Grefenstette, 2018], making the policy challenging to interpret for complex environments. Moreover, the function of *explainability* is absent, *i.e.* the agent cannot explain the importance of each input on its decision. Explainable agents should adaptively produce different explanations given different input states. A question thus arises: *How can we build interpretable and explainable RL agents that are robust to environmental changes?*

To this end, we introduce Neurally gUided Differentiable loGic policiEs (NUDGE), illustrated in Figure 1, that embody the advantages of logic: they are easily adaptable to environmental changes, composable, *interpretable* and *explainable* (because of our differentiable logic module). Given an input state, NUDGE extracts entities and their relations, converting raw states to a logic representations. This probabilistic relational states are used to deduce actions, using differentiable forward reasoning [Evans and Grefenstette, 2018, Shindo et al., 2023a]. NUDGE produces a policy that is both *interpretable*, *i.e.* provides a policy as a set of weighted interpretable rules that can be read out by humans, and *explainable*, *i.e.* explains which input is important using gradient-based attribution methods [Sundararajan et al., 2017] over logical representations.

To achieve an efficient learning with logic policies, we provide an algorithm to train NUDGE agents based on the PPO actor-critic framework. Moreover, we propose a novel rule-learning approach, called *Neurally-Guided Symbolic Abstraction*, where the candidate rules for the logic-based agents are obtained efficiently by being guided by neural-based agents. NUDGE distillates abstract representations of neural policies in the form of logic rules. Rules are assigned with their weights, and we perform gradient-based optimization using the PPO actor-critic framework.

Overall, we make the following contributions:

1. We propose NUDGE[3]: differentiable logical policies that learn interpretable rules and produce explanations for their decisions in complex environments. NUDGE uses neurally-guided symbolic abstraction to efficiently find a promising ruleset using pretrained neural-based agents guidance.

2. We empirically show that NUDGE agents: (i) can compete with neural-based agents, (ii) adapt to environmental changes, and (iii) are interpretable and explainable, *i.e.* produce interpretable policies as sets of weighted rules and provide explanations for their action selections.

3. We evaluate NUDGE on 2 classic Atari games and on 3 proposed object-centric logically challenging environments, where agents need relational reasoning in dynamic game-playing scenarios.

We start off by introducing the necessary background. Then we explain NUDGE's inner workings and present our experimental evaluation. Before concluding, we touch upon related work.

## 2 Background

We now describe the necessary background before formally introducing our NUDGE method.

---

[3]Code publicly available: `https://github.com/k4ntz/NUDGE`.

**Deep Reinforcement Learning**. In RL, the task is modelled as a Markov decision process, $\mathcal{M} = <\mathcal{S}, \mathcal{A}, P, R>$, where, at every timestep $t$, an agent in a state $s_t \in \mathcal{S}$, takes action $a_t \in \mathcal{A}$, receives a reward $r_t = R(s_t, a_t)$ and a transition to the next state $s_{t+1}$, according to environment dynamics $P(s_{t+1}|s_t, a_t)$. Deep agents attempt to learn a parametric policy, $\pi_\theta(a_t|s_t)$, in order to maximize the return (*i.e.* $\sum_t \gamma^t r_t$, with $\gamma \in [0, 1]$). The desired input to output (*i.e.* state to action) distribution is not directly accessible, as RL agents only observe returns. The value $V_{\pi_\theta}(s_t)$ (resp. Q-value $Q_{\pi_\theta}(s_t, a_t)$) function provides the return of the state (resp. state/action pair) following the policy $\pi_\theta$. Policy-based methods directly optimize $\pi_\theta$ using the noisy return signal, leading to potentially unstable learning. Value-based methods learn to approximate the value functions $\hat{V}_\phi$ or $\hat{Q}_\phi$, and implicitly encode the policy, *e.g.* by selecting the actions with the highest Q-value with a high probability [Mnih et al., 2015]. To reduce the variance of the estimated Q-value function, one can learn the advantage function $\hat{A}_\phi(s_t, a_t) = \hat{Q}_\phi(s_t, a_t) - \hat{V}_\phi(s_t)$. An estimate of the advantage function can be computed as $\hat{A}_\phi(s_t, a_t) = \sum_{i=0}^{k-1} \gamma^i r_{t+i} + \gamma^k \hat{V}_\phi(s_{t+k}) - \hat{V}_\phi(s_t)$ [Mnih et al., 2016]. The Advantage Actor-critic (A2C) methods both encode the policy $\pi_\theta$ (*i.e.* actor) and the advantage function $\hat{A}_\phi$ (*i.e.* critic), and use the critic to provide feedback to the actor, as in [Konda and Tsitsiklis, 1999]. To push $\pi_\theta$ to take actions that lead to higher returns, gradient ascent can be applied to $L^{PG}(\theta) = \hat{\mathbb{E}}[\log \pi_\theta(a \mid s) \hat{A}_\phi]$. Proximal Policy Optimization (PPO) algorithms ensure minor policy updates that avoid catastrophic drops [Schulman et al., 2017], and can be applied to actor-critic methods. To do so, the main objective constraints the policy ratio $r(\theta) = \frac{\pi_\theta(a|s)}{\pi_{\theta_{old}}(a|s)}$, following $L^{PR}(\theta) = \hat{\mathbb{E}}[\min(r(\theta)\hat{A}_\phi, \text{clip}(r(\theta), 1 - \epsilon, 1 + \epsilon)\hat{A}_\phi)]$, where clip constrains the input within $[1 - \epsilon, 1 + \epsilon]$. PPO actor-critic algorithm's global objective is $L(\theta, \phi) = \hat{\mathbb{E}}[L^{PR}(\theta) - c_1 L^{VF}(\phi)]$, with $L^{VF}(\phi) = (\hat{V}_\phi(s_t) - V(s_t))^2$ being the value function loss. An entropy term can also be added to this objective to encourage exploration.

**First-Order Logic (FOL).** In FOL, a *Language* $\mathcal{L}$ is a tuple $(\mathcal{P}, \mathcal{D}, \mathcal{F}, \mathcal{V})$, where $\mathcal{P}$ is a set of predicates, $\mathcal{D}$ a set of constants, $\mathcal{F}$ a set of function symbols (functors), and $\mathcal{V}$ a set of variables. A *term* is either a constant (*e.g.* `obj1`, `agent`), a variable (*e.g.* `O1`), or a term which consists of a function symbol. An *atom* is a formula $p(t_1, \ldots, t_n)$, where p is a predicate symbol (*e.g.* `closeby`) and $t_1, \ldots, t_n$ are terms. A *ground atom* or simply a *fact* is an atom with no variables (*e.g.* `closeby(obj1, obj2)`). A *literal* is an atom ($A$) or its negation ($\neg A$). A *clause* is a finite disjunction ($\vee$) of literals. A *ground clause* is a clause with no variables. A *definite clause* is a clause with exactly one positive literal. If $A, B_1, \ldots, B_n$ are atoms, then $A \vee \neg B_1 \vee \ldots \vee \neg B_n$ is a definite clause. We write definite clauses in the form of $A$ :- $B_1, \ldots, B_n$. Atom $A$ is called the *head*, and set of negative atoms $\{B_1, \ldots, B_n\}$ is called the *body*. We call definite clauses as *rules* for simplicity in this paper.

**Differentiable Forward Reasoning** is a data-driven approach of reasoning in FOL [Russell and Norvig, 2010]. In forward reasoning, given a set of facts and a set of rules, new facts are deduced by applying the rules to the facts. Differentiable forward reasoning [Evans and Grefenstette, 2018, Shindo et al., 2023a] is a differentiable implementation of the forward reasoning with tensor-based differentiable operations.

# 3 Neurally Guided Logic Policies

Figure 2 illustrates an overview of RL on NUDGE. They consist of a *policy reasoning* module and a *policy learning* module. NUDGE performs end-to-end differentiable policy reasoning based on forward reasoning, which computes action distributions given input states. On top of the reasoning module, policies are learned using neurally-guided symbolic abstraction and an actor-critic framework.

## 3.1 Policy Reasoning: Selecting Actions using Differentiable Forward Reasoning.

To realize NUDGE, we introduce a language to describe actions and states in FOL. Using it, we introduce differentiable policy reasoning using forward chaining reasoning.

### 3.1.1 Logic Programs for Actions

In RL, *states* and *actions* are key components since the agent performs the fundamental iteration of observing the state and taking an action to maximize its expected return. To achieve an efficient computation on first-order logic in RL settings, we introduce a simple language suitable for reasoning about states and actions.

We split the predicates set $\mathcal{P}$ into two different sets, *i.e.* , action predicates ($\mathcal{P}_A$), which define the actions and state predicates ($\mathcal{P}_S$) used for the observed states. If an atom $A$ consists of an action predicate, $A$ is called an *action atom*. If $A$ consists of a state predicate, $A$ is called *state atom*.

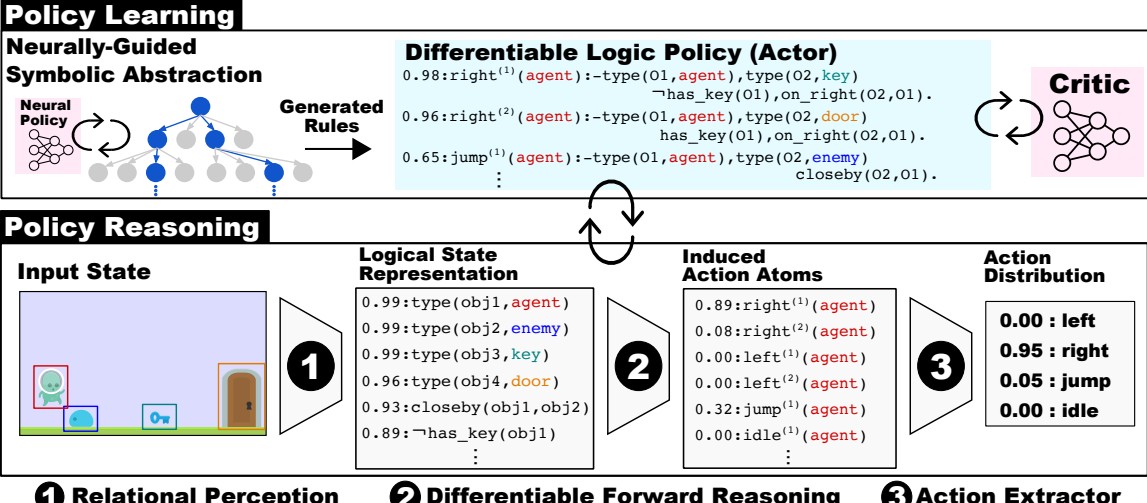

Figure 2: **NUDGE-RL**. **Policy Reasoning (bottom):** NUDGE agents incorporate end-to-end *reasoning* architectures from raw input based on differentiable forward reasoning. In the reasoning step, **(1)** the raw input state is converted into a logical representation, *i.e.* a set of atoms with probabilities. **(2)** Differentiable forward reasoning is performed using weighted action rules. **(3)** The final distribution over actions is computed using the results of differentiable reasoning. **Policy Learning (top):** Using the guidance of a pretrained neural policy, a set of candidate action rules is searched by *neurally-guided symbolic abstraction*, where promising action rules are produced. Then, randomly initialized weights are assigned to the action rules and are optimized using the critic of an actor-critic agent.

**Definition 1** *Action-state Language is a tuple of $(\mathcal{P}_A, \mathcal{P}_S, \mathcal{D}, \mathcal{V})$, where $\mathcal{P}_A$ is a set of action predicates, $\mathcal{P}_S$ is a set of state predicates, $\mathcal{D}$ is a set of constants for entities, and $\mathcal{V}$ is a set of variables.*

For example, for *Getout* illustrated in Figures 1 and 2, actual actions are: **left**, **right**, **jump**, and **idle**. We define action predicates $\mathcal{P}_A = \{\texttt{left}^{(1)}, \texttt{left}^{(2)}, \texttt{right}^{(1)}, \texttt{right}^{(2)}, \texttt{jump}^{(1)}, \texttt{idle}^{(1)}, ...\}$ and state predicates $\mathcal{P}_S = \{\texttt{type}, \texttt{closeby}, ...\}$. To encode different reasons for a given game action, we can generate several action predicates (*e.g.* $\texttt{right}^{(1)}$ and $\texttt{right}^{(2)}$ for **right**). By using these predicates, we can compose action atoms, *e.g.* $\texttt{right}^{(1)}(\texttt{agent})$, and state atoms, *e.g.* $\texttt{type}(\texttt{obj1}, \texttt{agent})$. An action predicate can also be a state predicate, *e.g.* in multiplayer settings. Now, we define rules to describe actions in the action-state language.

**Definition 2** *Let $X_A$ be an action atom and $X_S^{(1)}, \ldots, X_S^{(n)}$ be state atoms. An action rule is a rule, written as $X_A \colon -X_S^{(1)}, \ldots, X_S^{(n)}$.*

For example, for action **right**, we define an action rule as:

$$\texttt{right}^{(1)}(\texttt{agent}) \colon -\texttt{type}(\texttt{O1}, \texttt{agent}), \texttt{type}(\texttt{O2}, \texttt{key}), \neg\texttt{has\_key}(\texttt{O1}), \texttt{on\_right}(\texttt{O2}, \texttt{O1}).$$

which can be interpreted as *"The agent should go right if the agent does not have the key and the key is located on the right of the agent."*. Having several action predicates for an actual action (in the game) allows our agents to define several action rules that describe different reasons for the action.

### 3.1.2 Differentiable Logic Policies

We denote the set of actual actions by $\mathcal{A}$, the set of action rules by $\mathcal{C}$, the set of all of the facts by $\mathcal{G} = \mathcal{G}_A \cup \mathcal{G}_S$ where $\mathcal{G}_A$ is a set of action atoms and $\mathcal{G}_S$ is a set of state atoms. $\mathcal{G}$ contains all of the facts produced by a given FOL language. We here consider ordered sets, *i.e.* each element has its index. We also denote the size of the sets as: $A = |\mathcal{A}|$, $C = |\mathcal{C}|$, $G = |\mathcal{G}|$, and $G_A = |\mathcal{G}_A|$.

We propose *Differentiable Logic Policies*, which perform differentiable forward reasoning on action rules to produce a probability distribution over actions. The policy computation consists of 3 components: the relational perception module (1), the differentiable forward-reasoning module (2), and the action-extraction module (3).

The policy $\pi_{(\mathcal{C},\mathbf{W})}$ parameterized by a set of action rules $\mathcal{C}$ and the rules' weights $\mathbf{W}$ is computed as follows:

$$\pi_{(\mathcal{C},\mathbf{W})}(s_t) = p(a_t|s_t) = f^{act}\left(f^{reason}_{(\mathcal{C},\mathbf{W})}\left(f^{perceive}_{\Theta}(s_t)\right)\right), \tag{1}$$

with $f^{perceive}_{\Theta}: \mathbb{R}^N \to [0,1]^G$ a perception function that maps the raw input state $s_t \in \mathbb{R}^N$ into a set of probabilistic atoms, $f^{reason}_{(\mathcal{C},\mathbf{W})}: [0,1]^G \to [0,1]^{G_A}$ a differentiable forward reasoning function parameterized by a set of rules $\mathcal{C}$ and rule weights $\mathbf{W}$, and $f^{act}: [0,1]^{G_A} \to [0,1]^A$ an action-selection function, which computes the probability distribution over the action space.

**Relational Perception.** NUDGE agents take an object-centric state representations as input, obtained by *e.g.* using object detection [Redmon et al., 2016] or discovery [Lin et al., 2020, Delfosse et al., 2022] methods. These models return the detected objects and their attributes (*e.g.* class and positions). They are then converted into a probabilistic logic form with their relations, *i.e.* a set of facts with their probabilities. An input state $s_t \in \mathbb{R}^N$ is converted to a *valuation vector* $\mathbf{v} \in [0,1]^G$, which maps each fact to a probabilistic value. For example, let $\mathcal{G} = \{\text{type}(\text{obj1}, \text{agent}), \text{type}(\text{obj2}, \text{enemy}), \text{closeby}(\text{obj1}, \text{obj2}), \text{jump}(\text{agent})\}$. A valuation vector $[0.8, 0.6, 0.3, 0.0]^\top$ maps each fact to a corresponding probabilistic value. NUDGE performs differentiable forward reasoning by updating the initial valuation vector $\mathbf{v}^{(0)}$ $T$ times to obtain $\mathbf{v}^{(T)}$.

Initial valuation vector $\mathbf{v}^{(0)}$ is computed as follows. For each ground state atom $\text{p}(\text{t}_1, \ldots, \text{t}_\text{n}) \in \mathcal{G}_S$, *e.g.* $\text{closeby}(\text{obj1}, \text{obj2})$, a differentiable function is called to compute its probability, which maps each term $\text{t}_1, \ldots, \text{t}_\text{n}$ to vector representations according to the interpretation, *e.g.* $\text{obj1}$ and $\text{obj2}$ are mapped to their positions, then perform binary classification using the distance between them. For action atoms, zero is assigned as its initial probability (*e.g.* for $\text{jump}^{(1)}(\text{agent})$).

**Differentiable Forward Reasoning.** Given a set of candidate action rules $\mathcal{C}$, we create the reasoning function $f^{reason}_{(\mathcal{C},\mathbf{W})}: [0,1]^G \to [0,1]^{G_A}$, which takes the initial valuation vector and induces action atoms using weighted action rules. We assign weights to the action rules of $\mathcal{C}$ as follows: We fix the target programs' size, $M$, *i.e.* select $M$ rules out of $C$ candidate action rules. To do so, we introduce $C$-dimensional weights $\mathbf{W} = [\mathbf{w}_1, \ldots, \mathbf{w}_M]$ where $\mathbf{w}_i \in \mathbb{R}^C$ (*cf.* Figure 6 in the appendix). We take the *softmax* of each weight vector $\mathbf{w}_i \in \mathbf{W}$ to select $M$ action rules in a differentiable manner.

We perform $T$-step forward reasoning using action rules $\mathcal{C}$ with weights $\mathbf{W}$. We compose the differentiable forward reasoning function following Shindo et al. [2023a]. It computes soft logical entailment based on efficient tensor operations. Our differentiable forward reasoning module computes new valuation $\mathbf{v}^{(T)}$ including all induced atoms given weighted action rules $(\mathcal{C}, \mathbf{W})$ and initial valuation $\mathbf{v}^{(0)}$. Finally, we compute valuations on action atoms $\mathbf{v}_A \in [0,1]^{G_A}$ by extracting relevant values from $\mathbf{v}^{(T)}$. We provide details in App. E.

**Compute Action Probability.** Given valuations on action atoms $\mathbf{v}_A$, we compute the action distribution for actual actions. Let $\mathbf{a}_i \in \mathcal{A}$ be an actual action, and $v'_1, \ldots, v'_n \in \mathbf{v}_A$ be valuations which are relevant for $\mathbf{a}_i$ (*e.g.* valuations of $\text{right}^{(1)}(\text{agent})$ and $\text{right}^{(2)}(\text{agent})$ in $\mathbf{v}_A$ for the action **right**). We assign scores to each action $\mathbf{a}_i$ based on the *log-sum-exp* approach of Cuturi and Blondel [2017]: $val(\mathbf{a}_i) = \gamma \log \sum_{1 \le i \le n} \exp(v'_i/\gamma)$, that smoothly approximates the maximum value of $\{v'_1, \ldots, v'_n\}$. We use $\gamma > 0$ as a smoothing parameter. The action distribution is then obtained by taking the *softmax* over the evaluations of all actions.

## 3.2 Policy Learning

So far, we have considered that candidate rules for the policy are given, requiring human experts to handcraft potential rules. To avoid this, template-based rule generation [Evans and Grefenstette, 2018, Jiang and Luo, 2019] can be applied, but the number of generated rules increases exponentially with the number of entities and their potential relations. This technique is thus difficult to apply to complex environments where the agents need to reason about many different relations of entities.

To mitigate this problem, we propose an efficient learning algorithm for NUDGE that consists of *two* steps: neurally-guided symbolic abstraction and gradient-based optimization. First, NUDGE obtains a symbolic abstract set of rules, aligned with a given neural policy. The set of candidate rules is selected by neurally-guided top-$k$ search, *i.e.*, we generate a set of promising rules using a neural policy as an oracle. Then we assign randomized weights to each selected rule and perform differentiable reasoning. We finally optimize the rule weights using a "logic actor - neural critic" algorithm that aims at maximizing the expected return. Let us elaborate on each step.

### 3.2.1 Neurally Guided Symbolic Abstraction

Given a well-performing neural policy $\pi_\theta$, promising action rules for an RL task entail the same actions as the ones selected by $\pi_\theta$. We generate such rules by performing top-$k$ search-based abstraction, which uses the neural policy to evaluate rules efficiently. The inputs are initial rules $\mathcal{C}_0$, neural policy $\pi_\theta$. We start with elementary action rules and refine them to generate better action rules. $\mathcal{C}_{to\_open}$ is a set of rules to be refined, and initialized as $\mathcal{C}_0$. For each rule $C_i \in \mathcal{C}_{to\_open}$, we generate new rules by refining them as follows. Let $C_i = X_A \leftarrow X_S^{(1)}, \ldots, X_S^{(n)}$ be an already selected general action rule. Using a randomly picked ground or non-ground state atom $Y_S$ ($\neq X_S^{(i)}\ \forall i \in [1, ..., n]$), we refine the selected rule by adding a new state atom to its body, obtaining: $X_A \leftarrow X_S^{(1)}, \ldots X_S^{(n)}, Y_S$.

We evaluate each newly generated rule to select promising rules. We use the neural policy $\pi_\theta$ as a guide for the rule evaluation, *i.e.* rules that entail the same action as the neural policy $\pi_\theta$ are promising action rules. Let $\mathcal{X}$ be a set of states. Then we evaluate the rule $R$, following:

$$eval(R, \pi_\theta, \mathcal{X}) = \frac{1}{N(R, \mathcal{X})} \sum_{s \in \mathcal{X}} \pi_\theta(s)^\top \cdot \pi_{(\mathcal{R}, \mathbf{1})}(s), \tag{2}$$

where $N(R, \mathcal{X})$ is a normalization term, $\pi_{\mathcal{R}, \mathbf{1}}$ is the differentiable logic policy with rules $\mathcal{R} = \{R\}$ and rule weights $\mathbf{1}$, which is an $1 \times 1$ identity matrix (for consistent notation), and $\cdot$ is the dot product. Intuitively, $\pi_{(\mathcal{R}, \mathbf{1})}$ is the logic policy that has $R$ as its only action rule. If $\pi_{(\mathcal{R}, \mathbf{1})}$ produces a similar action distribution as the one produced by $\pi_\theta$, we regard the rule $R$ as a promising rule. We compute similarity scores between the neural policy $\pi_\theta$ and the logic one $\pi_{(\mathcal{R}, \mathbf{1})}$ using the dot product between the two action distributions, and average them across the states of $\mathcal{X}$. The normalization term avoids high scores for simple rules.

To compute the normalization term, we ground $R$, *i.e.* we replace variables with ground terms. We consider all of the possible groundings. Let $\mathcal{T}$ be the set of all of the possible variables substitutions to ground $R$. For each $\tau \in \mathcal{T}$, we get a ground rule $R\tau = X_A\tau \text{:-} X_S^{(1)}\tau, \ldots, X_S^{(n)}\tau$, where $X\tau$ represents the result of applying substitution $\tau$ to atom $X$. Let $\mathcal{J} = \{j_1, \ldots, j_n\}$ be indices of the ground atoms $X_S^{(1)}\tau, \ldots, X_S^{(n)}\tau$ in ordered set of ground atoms $\mathcal{G}$. Then, the normalization term is computed as:

$$N(R, \mathcal{X}) = \sum_{\tau \in \mathcal{T}} \sum_{s \in \mathcal{X}} \prod_{j \in \mathcal{J}} = \mathbf{v}_s^{(0)}[j], \tag{3}$$

where $\mathbf{v}_s^{(0)}$ is an initial valuation vector for state $s$, *i.e.* $f_\Theta^{perceive}(s)$. Eq. 3 quantifies how often the body atoms of the ground rule $R\tau$ are activated on the given set of states $\mathcal{X}$. Simple rules with fewer atoms in their body tend to have large values, and thus their evaluation scores in Eq. 2 tend to be small. After scoring all of the new rules, NUDGE select top-$k$ rules to refine them in the next step. To this end, all of the top-$k$ rules in each step will be returned as the candidate ruleset $\mathcal{C}$ for the policy (*cf.* App. A for more details about our algorithm).

We perform the action-rule generation for each action. In practice, NUDGE maintains the cached history $\mathcal{H}$ of states and actions produced by the neural policy. For a given action, the search quality increases together with the amount of times the action was selected, *i.e.* if $\mathcal{H}$ does not contain any record of the action, then all action rules (with this action as head) would get the same scores, leading to a random search.

NUDGE has thus produced candidate action rules $\mathcal{C}$, that will be associated with $\mathbf{W}$ to form untrained differentiable logic policy $\pi_{(\mathcal{C}, \mathbf{W})}$, as the ones described in Section 3.1.2.

### 3.2.2 Learning the Rules' Weights using the Actor-critic Algorithm

In the following, we consider a (potentially pretrained) actor-critic neural agent, with $v_\phi$ its differentiable state-value function parameterized by $\phi$ (critic). Given a set of action rules $\mathcal{C}$, let $\pi_{(\mathcal{C}, \mathbf{W})}$ be a differentiable logic policy. NUDGE learns the weights of the action rules in the following steps. For each non-terminal state $s_t$ of each episode, we store the actions sampled from the policy ($a_t \sim \pi_{(\mathcal{C}, \mathbf{W})}(s_t)$) and the next states $s_{t+1}$. We update the value function and the policy as follows:

$$\delta = r + \gamma v_\phi(s_{t+1}) - v_\phi(s_t) \tag{4}$$

$$\phi = \phi + \delta \nabla_\phi v_\phi(s_t) \tag{5}$$

$$\mathbf{W} = \mathbf{W} + \delta \nabla_\mathbf{W} \ln \pi_{(\mathcal{C}, \mathbf{W})}(s_t). \tag{6}$$

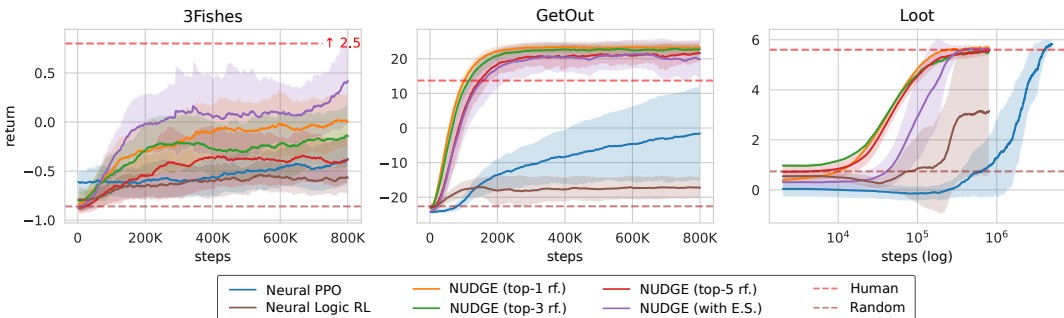

Figure 3: **NUDGE outperforms neural and logic baselines on the the $3$ logic environments.** Returns (avg.±std.) obtained by NUDGE, neural PPO and logic-based agents without abstraction through training. **NUDGE (Top-$k$ rf.)**, with $k \in \{1, 3, 10\}$ uses neurally-guided symbolic abstraction repeatedly until they get $k$ rules for each action predicate. **NUDGE (with E.S.)** uses rule set $\mathcal{C}$ supervised by an expert. **Neural Logic RL** composes logic-based policies by generating all possible rules without neurally-guided symbolic abstraction [Jiang and Luo, 2019]. Random and human baselines are also provided.

The logic policy $\pi_{(\mathcal{C}, \mathbf{W})}$ thus learn to maximize the expected return, potentially bootstrapped by the use of a pretrained neural critic. Moreover, to ease interpretability, NUDGE can prune the unused action rules (*i.e.* with low weights) by performing top-$k$ selection on the optimized rule weights after learning.

## 4 Experimental Evaluation

We here compare neural agents' performances to NUDGE ones, showcase NUDGE *interpretable* policies and its ability to report the importance of each input on their decisions, *i.e. explainable* logic policies. We use DQN agents (on Atari environments) and PPO actor-critic (on logic-ones) as neural baselines, for comparison and PPO as pretrained agents to guide the symbolic abstraction. All agent types receive object-centric descriptions of the environments. For clarity, we annotate action predicates on action rules with specific names on purpose, *e.g.* `right`$_{[\text{to\_key}]}$ instead of `right`$^{(1)}$ (when the rule describes an action **right** motivated to get the key).

We intend to compare agents with object-centric information bottlenecks. We thus had to extract object-centric states of the Atari environments. To do so, we make use of the Object-Centric Atari library [Delfosse et al., 2023]. As Atari games do not embed logic challenges, but are rather desgined to test the reflexes of human players, we also created 3 logic-oriented environments. We thus have modified environments from the Procgen [Mohanty et al., 2020] environments that are open-sourced along with our evaluation to have object-centric representations. Our environments are easily hackable. We provide variations of these environments also to evaluate the ease of adaptation of every agent type. In **GetOut**, the goal is to obtain a key, and then go to a door, while avoiding a moving enemy. **GetOut+** is a more complex variation with a larger world containing 5 enemies (among which 2 are static). In **3Fishes**, the agent controls a fish and is confronted with 2 other fishes, one smaller (that the agent needs to "eat", *i.e.* go to) and one bigger, that the agent needs to dodge. A variation is **3Fishes-C**, where the agent can eat green fishes and dodge red ones. Finally, in **Loot**, the agent can open 1 or 2 chests and their corresponding (*i.e.* same color) keys. In **Loot-C**, the chests have different colors. Further details and hyperparameters are provided in App. D.

We aim to answer the following research questions: **Q1.** How does NUDGE compare with neural and logic baselines? **Q2.** Can NUDGE agents easily adapt to environmental changes? **Q3.** Are NUDGE agents interpretable and explainable?

**NUDGE competes with existing methods (Q1).** We compare NUDGE with different baselines regarding their scores (or returns). First, we present scores obtained by trained DQN, Random and NUDGE agents (with expert supervision) on 2 Atari games (*cf.* Table 1). Our result show that NUDGE obtain better (Asterix) or similar (Freeway) scores than DQN. However, as said, Atari games are not logically challenging. We thus evaluate NUDGE on 3 logic environments. Figure 3 shows the returns in GetOut, 3Fishes, and Loot, with descriptions for each baseline in the caption. NUDGE obtains better performances than neural baselines (Neural PPO) on 3Fishes, is more stable on GetOut, *i.e.* less variance, and achieves faster convergence on Loot. This shows that NUDGE successfully distillates logic-based policies competing with neural baselines in different complex environments.

| Score ($\uparrow$) | Random | DQN | NUDGE |
|---|---|---|---|
| Asterix | 235 $\pm134$ | 124.5 | **6259** $\pm1150$ |
| Freeway | 0.0 $\pm0$ | **25.8** | 21.4 $\pm0.8$ |

| Score ($\uparrow$) | Random | Classic | NUDGE |
|---|---|---|---|
| GetOut | -22.5$\pm0.41$ | 11.59$\pm4.29$ | 17.86$\pm2.86$ |
| 3Fish | -0.73$\pm0.05$ | -0.24$\pm0.29$ | 0.05$\pm0.27$ |
| Loot | 0.57$\pm0.39$ | 0.51$\pm0.74$ | 5.66$\pm0.59$ |

Table 1: **Left: NUDGE agents can learn successful policies**. Trained NUDGE agents (with expert supervision) scores (avg. $\pm$ std.) on 2 OCAtari games [Delfosse et al., 2023]. Random and DQN (from van Hasselt et al. [2016]) are also provided. **Right: NUDGE outperforms non-differentiable symbolic reasoning baselines.** Returns obtained by NUDGE, classic and random agents our the 3 logic environmnents.

We also evaluate agents with a baseline without symbolic abstraction, where candidate rules are generated not being guided by neural policies, *i.e.* accept all of the generated rules in the rule refinement steps. This setting corresponds to the template-based approach [Jiang and Luo, 2019], but we train the agents by the actor-critic method, while vanilla policy gradient [Williams, 1992] is employed in [Jiang and Luo, 2019]. For the no-abstraction baseline and NUDGE, we provide initial action rules with basic type information, *e.g.* $\texttt{jump}^{(1)}(\texttt{agent}):-\texttt{type}(\texttt{O1}, \texttt{agent}), \texttt{type}(\texttt{O2}, \texttt{enemy})$, for each action rule. For this baseline, we generate 5 rules for GetOut, 30 rules for 3Fishes, and 40 rules for Loot in total to define all of the actual actions. NUDGE agents with small $k$ tend to have less rules, *e.g.* 5 rules in Getout, 6 rules in 3Fishes, and 8 rules in Loot for the top-1 refinement version. In Figure 3, the no-abstraction baselines perform worse than neural PPO and NUDGE in each environment, even though they have much more rules in 3Fishes and Loot. NUDGE thus composes efficient logic-based policies using neurally-guided symbolic abstraction. In App. B.1, we visualize the transition of the distribution of the rule weights in the GetOut environment.

To demonstrate the necessity of differentiability within symbolic reasoning in continuous RL environments, we simulated classic (*i.e.* discrete logic) agents, that reuse NUDGE agents' set of action rules, but with all weights fixed to 1.0. Such agents can still play better than ones with pure symbolic reasoners, as they still use fuzzy (*i.e.* , continuous) concepts, such as $\texttt{closeby}$. In all our environments, NUDGE clearly outperforms the classic counterparts (*cf.* Table 1 right). This experiment shows the superiority of differentiable policy reasoning (embedding weighted action rules) over policy without weighted rules (classic).

To further reduce the gap between our environments and those from environments Cao et al. [2022], we have adapted our loot environment (Loot-hard). In this environment, the agent first has to pick up keys and open their corresponding saves, before being able to exit the level (by going to a final exit tile). Our results in Table 2 (bottom), as well as curves on Figure 10 in the appendix show the dominance of NUDGE agents.

**NUDGE agents adapt to environment changes (Q2).**
We used the agents trained on the basic environment for this experimental evaluation, with no retraining or finetuning. For 3Fishes-C, we simply exchange the atom $\texttt{is\_bigger}$ with the atom $\texttt{same\_color}$. This easy modification is not applicable on the black-box networks of neural PPO agents. For GetOut+ and Loot-C, we do not apply any modification to the agents. Our results are summarized in Table 2. Note that the agents obtains better performances in the 3Fishes-C variation, dodging a (small) red fish is eas-

| Score ($\uparrow$) | 3Fishes-C | GetOut+ | Loot-C |
|---|---|---|---|
| Random | -0.6 $\pm0.2$ | -22.5 $\pm0.4$ | 0.6 $\pm0.3$ |
| Neural PPO | -0.4 $\pm0.1$ | -20.9 $\pm0.6$ | 0.8 $\pm0.5$ |
| NUDGE | **3.6** $\pm0.2$ | **3.6** $\pm3.0$ | **5.6** $\pm0.3$ |

Table 2: **NUDGE agents adapt to environmental changes and solve logically-challenging environments**. Returns obtained by NUDGE, neural PPO and random agents on our modified environments and on Loot-hard.

ier than a big one. For GetOut+, NUDGE performances have decreased as avoiding 5 enemies drastically increases the difficulty of the game. On Loot-C, the performances are similar to the ones obtained in the original game. Our experiments show that NUDGE logic agents can easily adapt to environmental changes.

**NUDGE agents are interpretable *and* explainable (Q3).** We show that NUDGE agents are interpretable and explainable by showing that (1) NUDGE produces interpretable policy as a set of weighted rules, and (2) NUDGE can show the importance of each atom, explaining its action choices.

The efficient neurally-guided learning on NUDGE enables the system to learn rules without inventing predicates with no specific interpretations, which are unavoidable in template-based approachs [Evans and Grefenstette, 2018, Jiang and Luo, 2019]. Thus, the policy can easily be read out by extracting action rules with high weights. Figure 5 shows some action rules discovered by NUDGE in GetOut. The first rule says: *"The agent should jump when the enemy is close to the agent (to avoid the enemy)."*. The produced NUDGE is an *interpretable* policy

```
0.57:jump(agent):-type(O1,agent),type(O2,enemy),closeby(O1,O2).
0.29:right[to_key](agent):-type(O1,agent),type(O2,key),on_right(O2,O1),¬has_key(O1).
0.56:right[to_door](agent):-type(O1,agent),type(O2,door),on_right(O2,O1),has_key(O1).
0.32:left[to_key](agent):- type(O1,agent),type(O2,key),on_right(O1,O2),¬has_key(O1).
0.30:left[to_door](agent):-type(O1,agent),type(O2,door),on_left(O2,O1),has_key(O1).
```

Figure 5: **NUDGE produces interpretable policies as sets of weighted rules.** A subset of the weighted action rules from the Getout environment. Full policies for every logic environment are provided in App. B.3.

with its set of weighted rules using interpretable predicates. For each state, we can also look at the valuation of each atom and the selected rule.

Moreover, contrary to classic logic policies, the differentiable ones produced by NUDGE allow us to compute the *attribution values* over the logical representations via backpropagation of the policies' gradients. We can compute the action gradients w.r.t. input atoms, *i.e.* $\partial \mathbf{v}_A / \partial \mathbf{v}^{(0)}$, as shown in Figure 4, which represent the relevance scores of the probabilistic input atoms $\mathbf{v}^{(0)}$ for the actions given a specific state. The explanation is computed on the state shown in Figure 1, where the agent takes **right** as its action. Important atoms receive large gradients, *e.g.* ¬has_key(agent) and on_right(obj2,obj1). By extracting relevant atoms with large gradients, NUDGE can produce clear explanations for the action selection. For example, by

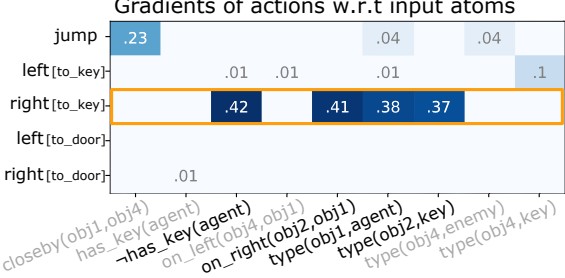

Figure 4: **Explanation using inputs' gradients**. The action gradient w.r.t. input atoms, *i.e.* $\partial \mathbf{v}_A / \partial \mathbf{v}^{(0)}$, on the state shown in Figure 1. **right** was selected, due to the highlighted relevant atoms (with large gradients).

extracting the atoms wrapped in orange in Figure 4, NUDGE can explain the motivation: *"The agent decides to go right because it does not have the key and the key is located on the right-side of it."*. These results show that NUDGE policies are *interpretable* and *explainable*, *i.e.* each action predicate is defined by rules that humans can fully understand, and gradient-based explanations for the action selections can be efficiently produced.

We also computed explanations of a neural agent on the same state, also using the input-gradients method. The concepts which received the highest explanation scores are: the key x-axis position (1.22), the key x-axis velocity (1.21), the enemy y-axis velocity (1.10), and the door x-axis position (0.77). For a consistent explanation, the agent x-axis position should also be among the highest scores. For this specific state, the neural agent seems to be placing its explanations on non relevant concepts (such as the key's velocity). Stammer et al. [2021] have shown that neural networks trained on logical concepts tend to produce wrong explanations often, and additional regularization/supervision about explanations during training is necessary to force neural networks to produce correct explanations. This approach requires additional efforts to label each state with its correct explanation. NUDGE policies produces better explanations than neural-based agents. More explanations produced by NUDGE are provided in Figure 11 in the appendix.

## 5 Related Work

Relational RL [Dzeroski et al., 2001, Kersting et al., 2004, Kersting and Driessens, 2008, Lang et al., 2012] has been developed to tackle RL tasks in relational domains. Relational RL frameworks incorporate logical representations and use probabilistic reasoning. With NUDGE, we make use of differentiable logic programming. The Neural Logic Reinforcement Learning (NLRL) [Jiang and Luo, 2019] framework is the first to integrate Differentiable Inductive Logic Programming (∂ILP) [Evans and Grefenstette, 2018] to RL domain. ∂ILP learns generalized logic rules from examples by gradient-based optimization. NLRL adopts ∂ILP as a policy function. We extend this approach by proposing neurally-guided

| | FOL | N.G. | Int. | Exp. |
|---|---|---|---|---|
| NLRL | ✓ | ✗ | ✓ | ✗ |
| NeSyRL | ✓ | ✗ | ✓ | ✗ |
| DiffSES | ✗ | ✓ | ✓ | ✗ |
| NUDGE | ✓ | ✓ | ✓ | ✓ |

Table 3: **Logic-based RL methods comparison**: First Order Logic (FOL), neurally-guided search (N.G.), interpretability (Int.), and explainability (Exp.).

symbolic abstraction embracing extensive work of ∂ILP [Shindo et al., 2021b,a, 2023b] for learning complex programs including functors in visual scenes, allowing agents to learn interpretable action rules efficiently for complex environments.

GALOIS [Cao et al., 2022] is a framework to represent policies as logic programs using the *sketch* setting [Solar-Lezama, 2008], where programs are learned to fill blanks. NUDGE performs structure learning from scratch using policy gradients. KoGun [Zhang et al., 2020] integrates human knowledge as a prior for RL agents. NUDGE learns a policy as a set of weighted rules and thus also can integrate human knowledge. Neuro-Symbolic RL (NeSyRL) [Kimura et al., 2021] uses Logical Neural Networks (LNNs) [Riegel et al., 2020] for the policy computation. LNNs parameterize the soft logical operators while NUDGE parameterizes rules with their weights. Deep Relational RL approaches [Zambaldi et al., 2018] achieve relational reasoning as a neural network, but NUDGE explicitly encodes relations in logic. Many languages for planning and RL tasks have been developed [Fikes and Nilsson, 1971, Fox and Long, 2003]. Our approach is inspired by *situation calculus* [Reiter, 2001], which is an established framework to describe states and actions in logic.

Symbolic programs within RL have been investigated, *e.g.* program guided agent [Sun et al., 2020], program synthesis [Zhu et al., 2019], PIRL [Verma et al., 2018], SDRL [Lyu et al., 2019], interpretable model-based hierarchical RL Xu and Fekri [2021], deep symbolic policy Landajuela et al. [2021], and DiffSES Zheng et al. [2021]. These approaches use domain specific languages or propositional logic, and address either of interpretability or explainability of RL. To this end, in Table 3, we compare NUDGE with the most relevant approaches that share at least 2 of the following aspects: supporting first-order logic, neural guidance, interetability and explainability. NUDGE is the first method to use neural guidance for differentiable first-order logic policies and to address both *interpretability* and *explainability*. Specifically, PIRL develops functional programs with neural guidance using *sketches*, which define a grammar of programs to be generated. In contrast, NUDGE performs structure learning from scratch using a neural guidance with a language bias of *mode declarations* [Muggleton, 1995, Cropper et al., 2022] restricting the search space. As NUDGE uses first-order logic, it can incorporate background knowledge in a declarative form, *e.g.* with a few lines of relational atoms and rules. To demonstrate this, NUDGE has been evaluated on challenging environments where the main interest is relational reasoning.

# 6 Conclusion

We proposed NUDGE, an interpretable and explainable policy reasoning and learning framework for reinforcement learning. NUDGE uses differentiable forward reasoning to obtain a set of interpretable weighted rules as policy. It performs neurally-guided symbolic abstraction, which efficiently distillates symbolic representations from a neural policy, and incorporate an actor-critic method to perform gradient-based policy optimization. We empirically demonstrated that NUDGE agents (1) can compete with neural based policies, (2) use logical representations to produce both interpretable and explainable policies and (3) can automatically adapt or easily be modified and are thus robust to environmental changes.

**Societal and environmental impact.** As NUDGE agents can explain the importance of each the input on their decisions, and as their rules are interpretable, it can help us understanding the decisions of RL agents trained in sensitive complicated domains, and potentially help discover biases and misalignments of discriminative nature. While the distillation of the learned policy into a logic ones augments the computational resources needed for completing the agents learning process, we believe that the logic the agents' abstraction capabilities will overall reduce the need of large neural networks, and will remove the necessity of retraining agents for each environmental change, leading to an overall reduction of necessary computational power.

**Limitation and Future Work.** NUDGE is only complete if provided with a sufficiently expressive language (in terms of predicates and entities) to approximate neural policies. Interesting lines of future research can include: (i) an automatic discovery of predicates, using *e.g.* predicate invention [Muggleton et al., 2015], (ii) augmenting the number of accessible entities to reason on. Explainable interactive learning [Teso and Kersting, 2019] in RL can integrate NUDGE agents, since it produces explanations on logical representations. Causal RL [Madumal et al., 2020] and meta learning [Mishra et al., 2018] also constitute interesting avenues for NUDGE's development. Finally, we could incorporate objects extraction methods [Delfosse et al., 2022] within our NUDGE agents, to obtain logic agents that extract object and their properties from images.

**Acknowledgements.** The authors thank the anonymous reviewers for their valuable feedback. This research work has been funded by the German Federal Ministry of Education and Research, the Hessian Ministry of Higher Education, Research, Science and the Arts (HMWK) within their joint support of the National Research Center for Applied Cybersecurity ATHENE, via the "SenPai: XReLeaS" project. We gratefully acknowledge support by the Federal Ministry of Education and Research (BMBF) AI lighthouse project "SPAICER" (01MK20015E), the EU ICT-48 Network of AI Research Excellence Center "TAILOR" (EU Horizon 2020, GA No 952215), and the Collaboration Lab "AI in Construction" (AICO).

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

# Supplemental Materials

## A Details on Neurally-Guided Symbolic Abstraction

We here provide details on the neurally-guided symbolic abstraction algorithm.

### A.1 Algorithm of Neurally-Guided Symbolic Abstraction

We show the algorithm of neurally-guided symbolic abstraction in Algorithm 1.

---
**Algorithm 1** *Neurally-Guided Symbolic Abstraction*

---
**Input:** $\mathcal{C}_0, \pi_\theta$, hyperparameters $(N_{beam}, T_{beam})$
1: $\mathcal{C}_{to\_open} \leftarrow \mathcal{C}_0$
2: $\mathcal{C} \leftarrow \emptyset$
3: $t = 0$
4: **while** $t < T_{beam}$ **do**
5:      $\mathcal{C}_{beam} \leftarrow \emptyset$
6:      **for** $C_i \in \mathcal{C}_{to\_open}$ **do**
7:          $\mathcal{C} = \mathcal{C} \cup \{C_i\}$
8:          **for** $R \in \rho(C_i)$ **do**
             `# Evaluate each clause`
9:              $score = eval(R, \pi_\theta)$
             `# select top-k rules`
10:             $\mathcal{C}_{beam} = top\_k(\mathcal{C}_{beam}, R, score, N_{beam})$
        `# selected rules are refined next`
11:      $\mathcal{C}_{to\_open} = \mathcal{C}_{beam}$
12:      $t = t + 1$
     **return** $\mathcal{C}$

---

### A.2 Rule Generation

At line 8 in Algorithm 1, given action rule $C$, we generate new action rules using the following refinement operation:

$$\rho(C) = \{X_A \leftarrow X_S^{(1)}, \dots X_S^{(n)}, Y_S \mid Y_S \in \mathcal{G}_S^* \wedge Y_S \neq X_S^{(i)}\}, \tag{7}$$

where $\mathcal{G}_S^*$ is a non-ground state atoms. This operation is a specification of *(downward) refinement operator*, which a fundamental technique for rule learning in ILP [Nienhuys-Cheng and de Wolf, 1997], for action rules to solve RL tasks.

We use mode declarations [Muggleton, 1995, Cropper et al., 2022] to define the search space, *i.e.* $\mathcal{G}_S^*$ in Eq. 7 which are defined as follows. A mode declaration is either a head declaration $\text{modeh}(r, p(\text{mdt}_1, \dots, \text{mdt}_n))$ or a body declaration $\text{modeb}(r, p(\text{mdt}_1, \dots, \text{mdt}_n))$, where $r \in \mathbb{N}$ is an integer, $p$ is a predicate, and $\text{mdt}_i$ is a mode datatype. A mode datatype is a tuple $(\text{pm}, \text{dt})$, where $\text{pm}$ is a place-marker and $\text{dt}$ is a datatype. A place-marker is either $\#$, which represents constants, or $+$ (resp. $-$), which represents input (resp. output) variables. $r$ represents the number of the usages of the predicate to compose a solution. Given a set of mode declarations, we can determine a finite set of rules to be generated by the rule refinement.

Now we describe mode declarations we used in our experiments. For Getout, we used the following mode declarations:

$$\text{modeb}(2, \text{type}(-\text{object}, +\text{type}))$$
$$\text{modeb}(1, \text{closeby}(+\text{object}, +\text{object}))$$
$$\text{modeb}(1, \text{on\_left}(+\text{object}, +\text{object}))$$
$$\text{modeb}(1, \text{on\_right}(+\text{object}, +\text{object}))$$
$$\text{modeb}(1, \text{have\_key}(+\text{object}))$$
$$\text{modeb}(1, \text{not\_have\_key}(+\text{object}))$$

For 3Fishes, we used the following mode declarations:

$$\text{modeb}(2, \text{type}(-\text{object}, +\text{type}))$$
$$\text{modeb}(1, \text{closeby}(+\text{object}, +\text{object}))$$
$$\text{modeb}(1, \text{on\_top}(+\text{object}, +\text{object}))$$
$$\text{modeb}(1, \text{at\_bottom}(+\text{object}, +\text{object}))$$
$$\text{modeb}(1, \text{on\_left}(+\text{object}, +\text{object}))$$
$$\text{modeb}(1, \text{on\_right}(+\text{object}, +\text{object}))$$
$$\text{modeb}(1, \text{bigger\_than}(+\text{object}, +\text{object}))$$
$$\text{modeb}(1, \text{high\_level}(+\text{object}, +\text{object}))$$
$$\text{modeb}(1, \text{low\_level}(+\text{object}, +\text{object}))$$

For Loot, we used the following mode declarations:

$$\text{modeb}(2, \text{type}(-\text{object}, +\text{type}))$$
$$\text{modeb}(2, \text{color}(+\text{object}, \#\text{color}))$$
$$\text{modeb}(1, \text{closeby}(+\text{object}, +\text{object}))$$
$$\text{modeb}(1, \text{on\_top}(+\text{object}, +\text{object}))$$
$$\text{modeb}(1, \text{at\_bottom}(+\text{object}, +\text{object}))$$
$$\text{modeb}(1, \text{on\_left}(+\text{object}, +\text{object}))$$
$$\text{modeb}(1, \text{on\_right}(+\text{object}, +\text{object}))$$
$$\text{modeb}(1, \text{have\_key}(+\text{object}))$$

# B  Additional Results

## B.1  Weights learning

Fig. 6 shows the NUDGE agent $\pi_{(\mathcal{C}, \mathbf{W})}$ parameterized by rules $\mathcal{C}$ and weights $\mathbf{W}$ before training (top) and after training (bottom) on the GetOut environment. Each element on the x-axis of the plots corresponds to an action rule. In this examples, we have 10 action rules $\mathcal{C} = \{C_0, C_1, \ldots, C_9\}$, and we assign $M = 5$ weights *i.e.* $\mathbf{W} = [\mathbf{w}_0, \mathbf{w}_1, \ldots, \mathbf{w}_4]$. The distributions of rule weighs with *softmax* are getting peaked by learning to maximize the return. The right 4 rules are redundant rules, and theses rules get low weights after learning.

## B.2  Deduction Pipeline

Fig. 7 provides the deduction pipeline of a NUDGE agent on 3 different states. Facts can be deduced from an object detection method, or directly given by the object centric environment. For state #1, the agent chooses to jump as the jump action is prioritized over the other ones and all atoms that compose this rules' body have high valuation (including closeby). In state #2, the agent chose to go **left** as the rule left_key is selected. In state #3, the agent selects **right** as the rule right_door has the highest forward chaining evaluation.

## B.3  Policies of every logic environment.

We show the logic policies obtained by NUDGE in GetOut, 3Fishes, and Loot in Fig. 8, *e.g.* the first line of GetOut, "$0.574 : \text{jump}(\text{X}){:}{-}\text{closeby}(\text{O1}, \text{O2}), \text{type}(\text{O1}, \text{agent}), \text{type}(\text{O2}, \text{enemy}).$", represents that the action rule is chosen by the weight vector $\mathbf{w}_1$ with a value $0.574$. NUDGE agents have several weight vectors $\mathbf{w}_1, \ldots, \mathbf{w}_M$ and thus several chosen action rules are shown for each environment.

# C  Illustrations of our environments

We showcase in Fig. 9 one state of the 3 object-centric environments and their variations. In **GetOut** (blue humanoid agent), the goal is to obtain a key, then go to a door, while avoiding a moving enemy. **GetOut-2En**

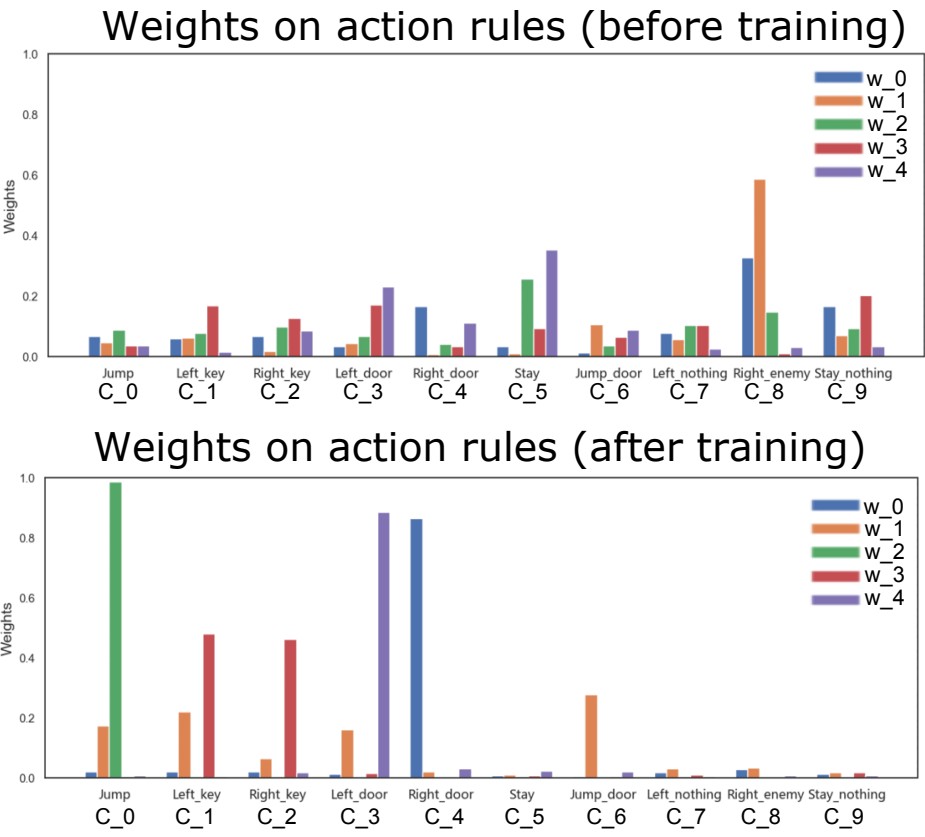

Figure 6: Weights on action rules via softmax before training (top) and after training (bottom) on NUDGE in GetOut. Each element on the x-axis of the plots corresponds to an action rule. NUDGE learns to get high returns while identifying useful action rules to solve the RL task. The right 5 rules are redundant rules, and theses rules get low weights after learning.

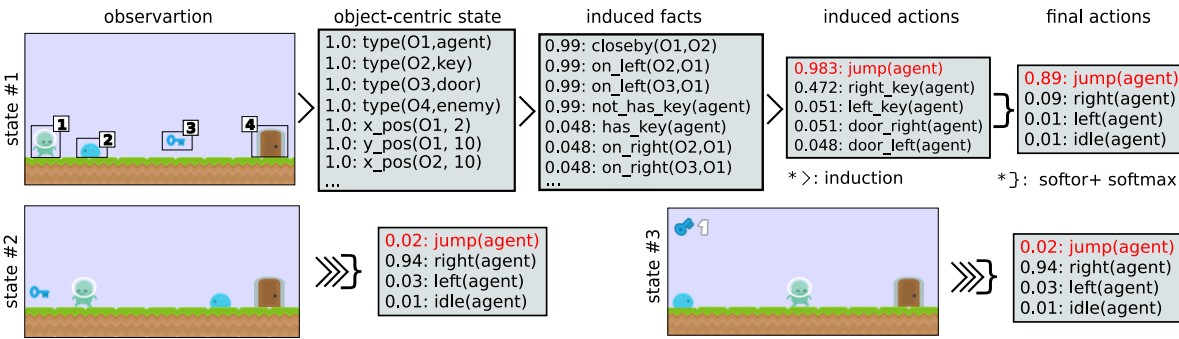

Figure 7: The logic reasoning of NUDGE agents makes them interpretable. The detailed logic pipeline for the input state #1 of the *Getout* environment and the condensed action selection for state #2 and state #3.

is a variation with 2 enemies. In **3Fishes**, the agent controls a green fish and is confronted with 2 other fishes, one smaller (that the agent need to "eat", *i.e.* go to) and one bigger, that the agent needs to dodge. A variation is **3Fishes-C**, where the agent can eat green fishes and dodge red ones, all fishes have the same size. Finally, in **Loot**, the (orange) agent is exposed with 1 or 2 chests and their corresponding (*i.e.* same color) keys. In **Loot-C**, the chests have different colors. All 3 environment are *stationary* in the sense of Delfosse et al. [2021].

```
# GetOut
0.574:jump(X):-closeby(O1,O2),type(O1,agent),type(O2,enemy).
0.315:left_go_get_key(X):-not_have_key(X),on_right(O1,O2),type(O1,agent),type(O2,key).
0.296:left_go_to_door(X):-have_key(X),on_left(O2,O1),type(O1,agent),type(O2,door).
0.291:right_go_get_key(X):-not_have_key(X),on_left(O1,O2),type(O1,agent),type(O2,key).
0.562:right_go_to_door(X):-have_key(X),on_left(O1,O2),type(O1,agent),type(O2,door).

#3Fishes
0.779:right_to_eat(X):-is_bigger_than(O1,O2),on_left(O2,O1),type(O1,agent),
                       type(O2,fish).
0.445:down_to_dodge(X):-is_bigger_than(O2,O1),on_left(O2,O1),type(O1,agent),
                        type(O2,fish).
0.579:down_to_eat(X):-high_level(O1,O2),is_smaller_than(O2,O1),type(O1,agent),
                      type(O2,fish).
0.699:up_to_dodge(X):-closeby(O2,O1),is_smaller_than(O1,O2),low_level(O2,O1),
                      type(O1,agent),type(O2,fish).
0.601:up_to_eat(X):-is_bigger_than(O2,O1),on_left(O2,O1),type(O1,agent),
                    type(O2,fish).
0.581:left_to_eat(X):-closeby(O1,O2),on_right(O1,O2),type(O1,agent),type(O2,fish).

# Loot
0.844:up_to_door(X):-close(O1,O2),have_key(O2),on_top(O2,O1),type(O1,agent),
                     type(O2,door).
0.268:right_to_key(X):-close(O1,O2),on_right(O2,O1),type(O1,agent),type(O2,key).
0.732:right_to_door(X):-close(O1,O2),have_key(O2),on_left(O1,O2),type(O1,agent),
                        type(O2,door).
0.508:up_to_key(X):-close(O1,O2),on_top(O2,O1),type(O1,agent),type(O2,key).
0.995:left_to_door(X):-close(O1,O2),have_key(O2),on_left(O2,O1),type(O1,agent),
                       type(O2,door).
0.414:down_to_key(X):-close(O1,O2),on_top(O1,O2),type(O1,agent),type(O2,key).
0.992:down_to_door(X):-close(O1,O2),have_key(O2),on_top(O1,O2),type(O1,agent),
                       type(O2,door).
0.447:left_to_key(X):-close(O1,O2),on_left(O2,O1),type(O1,agent),type(O2,key).
```

Figure 8: **NUDGE produces an interpretable policy as set of weighted rules.** Weighted action rules discovered by NUDGE in the each logic environment.

## D    Hyperparameters and rules sets

### D.1    Hyperparameters

We here provide the hyperparameters used in our experiments. We set the clip parameter $\epsilon_{clip} = 0.2$, the discount factor $\gamma = 0.99$. We use the Adam optimizer, with $1e-3$ as actor learning rate, $3e-4$ as critic learning rate. The episode length is $500$ timesteps. The policy is updated every $1000$ steps We train every algorithm for $800k$ steps on each environment, apart from neural PPO, that needed $5M$ steps on Loot. We use an epsilon greedy strategy with $\epsilon = max(e^{\frac{-episode}{500}}, 0.02)$.

### D.2    Rules set

All the rules set $\mathcal{C}$ of the different NUDGE and logic agents are available at `https://github.com/k4ntz/NUDGE` in the folder `nsfr/nsfr/data/lang`.

## E    Details of Differentiable Forward Reasoning

We provide details of differentiable forward reasoning used in NUDGE. We denote a valuation vector at time step $t$ as $\mathbf{v}^{(t)} \in [0, 1]^G$. We also denote the $i$-th element of vector $\mathbf{x}$ by $\mathbf{x}[i]$, and the $(i, j)$-th element of matrix $\mathbf{X}$ by $\mathbf{X}[i, j]$. The same applies to higher dimensional tensors.

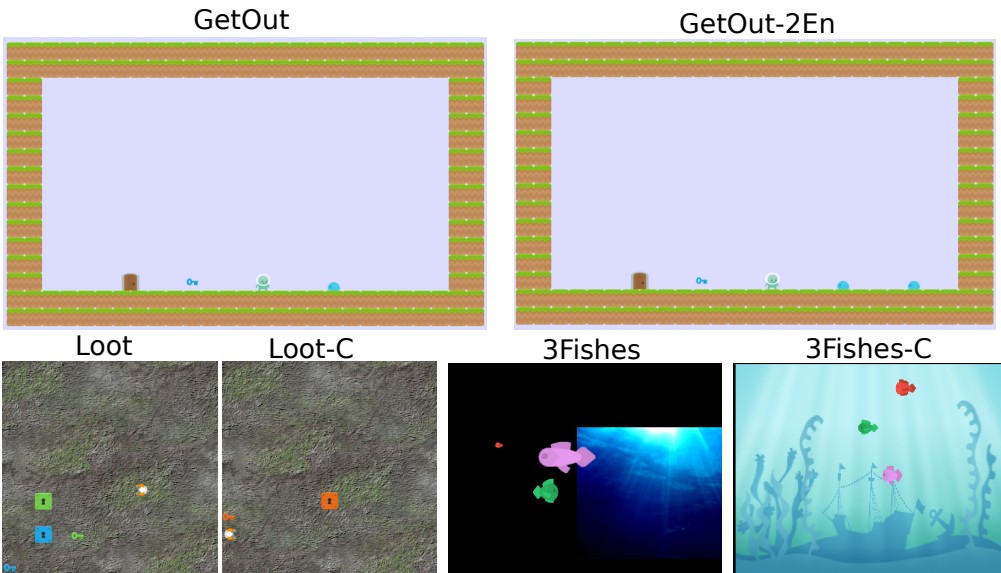

Figure 9: Pictures of our environments (**GetOut**, **Loot** and **3Fishes**) and their variations (**GetOut-2En**, **Loot-C** and **3Fishes-C**). All these environments can provide object-centric state descriptions (instead of pixel-based states).

### E.1 Differentiable Forward Reasoning

We compose the reasoning function $f^{reason}_{(\mathcal{C},\mathbf{W})} : [0,1]^G \to [0,1]^{G_A}$, which takes the initial valuation vector and returns valuation vector for induced action atoms. We describe each step in detail.

**(Step 1) Encode Logic Programs to Tensors.** To achieve differentiable forward reasoning, each action rule is encoded to a tensor representation. Let $S$ be the number of the maximum number of substitutions for existentially quantified variables in $\mathcal{C}$, and $L$ be the maximum length of the body of rules in $\mathcal{C}$. Each action rule $C_i \in \mathcal{C}$ is encoded to a tensor $\mathbf{I}_i \in \mathbb{N}^{G \times S \times L}$, which contain the indices of body atoms. Intuitively, $\mathbf{I}_i[j,k,l]$ is the index of the $l$-th fact (subgoal) in the body of the $i$-th rule to derive the $j$-th fact with the $k$-th substitution for existentially quantified variables.

For example. let $R_0 = \texttt{jump(agent):-type(O1,agent),type(O2,enemy),closeby(O1,O2)} \in \mathcal{C}$ and $F_2 = \texttt{jump(agent)} \in \mathcal{G}$, and we assume that constants for objects are $\{\texttt{obj1},\texttt{obj2}\}$. $R_0$ has existentially quantified variables $\texttt{O1}$ and $\texttt{O2}$ on the body, so we obtain ground rules by substituting constants. By considering the possible substitutions for $\texttt{O1}$ and $\texttt{O2}$, namely $\{\texttt{O1/obj1},\texttt{O2/obj2}\}$ and $\{\texttt{O1/obj2},\texttt{O2/obj1}\}$, we have *two* ground rules, as shown in top of Table 4. Bottom rows of Table 4 shows elements of tensor $\mathbf{I}_{0,:,0,:}$ and $\mathbf{I}_{0,:,1,:}$. Facts $\mathcal{G}$ and the indices are represented on the upper rows in the table. For example, $\mathbf{I}_{0,2,0,:} = [3,6,7]$ because $R_0$ entails $\texttt{jump(agent)}$ with the first ($k = 0$) substitution $\tau = \{\texttt{O1/obj1},\texttt{O2/obj2}\}$. Then the subgoal atoms are $\{\texttt{type(obj1,agent)},\texttt{type(obj2,enemy)},\texttt{closeby(obj1,obj2)}\}$, which have indices $[3,6,7]$, respectively. The atoms which have a different predicate, e.g., $\texttt{closeby(obj1,obj2)}$, will never be entailed by clause $R_0$. Therefore, the corresponding values are filled with 0, which represents the index of the *false* atom.

**(Step 2) Assign Rule Weights.** We assign weights to compose the policy with several action rules as follows: (i) We fix the target programs' size as $M$, *i.e.* , where we try to find a policy with $M$ action rules. (ii) We introduce $C$-dim weights $\mathbf{W} = [\mathbf{w}_1,\ldots,\mathbf{w}_M]$. (iii) We take the *softmax* of each weight vector $\mathbf{w}_j \in \mathbf{W}$ and softly choose $M$ action rules out of $C$ action rules to compose the policy.

**(Step 3) Perform Differentiable Inference.** We compute 1-step forward reasoning using weighted action rules, then we recursively perform reasoning to compute $T$-step reasoning.

$$(k = 0) \quad \texttt{jump(agent):-type(obj1,agent),type(obj2,enemy),closeby(obj1,obj2).}$$
$$(k = 1) \quad \texttt{jump(agent):-type(obj2,agent),type(obj1,enemy),closeby(obj2,obj1).}$$

| $j$ | 0 | 1 | 2 | 3 | 4 | 5 |
|---|---|---|---|---|---|---|
| $\mathcal{G}$ | $\perp$ | $\top$ | $\texttt{jump(agent)}$ | $\texttt{type(obj1,agent)}$ | $\texttt{type(obj2,agent)}$ | $\texttt{type(obj1,enemy)}$ |
| $\mathbf{I}_{0,j,0,:}$ | $[0,0,0]$ | $[1,1,1]$ | $[3,6,7]$ | $[0,0,0]$ | $[0,0,0]$ | $[0,0,0]$ |
| $\mathbf{I}_{0,j,1,:}$ | $[0,0,0]$ | $[1,1,1]$ | $[4,5,8]$ | $[0,0,0]$ | $[0,0,0]$ | $[0,0,0]$ |

| $j$ | 6 | 7 | 8 | $\dots$ |
|---|---|---|---|---|
| $\mathcal{G}$ | $\texttt{type(obj2,enemy)}$ | $\texttt{closeby(obj1,obj2)}$ | $\texttt{closeby(obj2,obj1)}$ | $\dots$ |
| $\mathbf{I}_{0,j,0,:}$ | $[0,0,0]$ | $[0,0,0]$ | $[0,0,0]$ | $\dots$ |
| $\mathbf{I}_{0,j,1,:}$ | $[0,0,0]$ | $[0,0,0]$ | $[0,0,0]$ | $\dots$ |

Table 4: Example of ground rules (top) and elements in the index tensor (bottom). Each fact has its index, and the index tensor contains the indices of the facts to compute forward inferences.

**[(i) Reasoning using an action rule]** First, for each action rule $C_i \in \mathcal{C}$, we evaluate body atoms for different grounding of $C_i$ by computing $b_{i,j,k}^{(t)} \in [0,1]$:

$$b_{i,j,k}^{(t)} = \prod_{1 \leq l \leq L} \mathbf{gather}(\mathbf{v}^{(t)}, \mathbf{I}_i)[j,k,l] \tag{8}$$

where $\mathbf{gather} : [0,1]^G \times \mathbb{N}^{G \times S \times L} \to [0,1]^{G \times S \times L}$ is:

$$\mathbf{gather}(\mathbf{x}, \mathbf{Y})[j,k,l] = \mathbf{x}[\mathbf{Y}[j,k,l]]. \tag{9}$$

The $\mathbf{gather}$ function replaces the indices of the body state atoms by the current valuation values in $\mathbf{v}^{(t)}$. To take logical *and* across the subgoals in the body, we take the product across valuations. $b_{i,j,k}^{(t)}$ represents the valuation of body atoms for $i$-th rule using $k$-th substitution for the existentially quantified variables to deduce $j$-th fact at time $t$.

Now we take logical *or* softly to combine all of the different grounding for $C_i$ by computing $c_{i,j}^{(t)} \in [0,1]$:

$$c_{i,j}^{(t)} = softor^\gamma(b_{i,j,1}^{(t)}, \dots, b_{i,j,S}^{(t)}) \tag{10}$$

where $softor^\gamma$ is a smooth logical *or* function:

$$softor^\gamma(x_1, \dots, x_n) = \gamma \log \sum_{1 \leq i \leq n} \exp(x_i/\gamma), \tag{11}$$

where $\gamma > 0$ is a smooth parameter. Eq. 11 is an approximation of the *max* function over probabilistic values based on the *log-sum-exp* approach [Cuturi and Blondel, 2017].

**[(ii) Combine results from different action rules]** Now we apply different action rules using the assigned weights by computing $h_{j,m}^{(t)} \in [0,1]$:

$$h_{j,m}^{(t)} = \sum_{1 \leq i \leq C} w_{m,i}^* \cdot c_{i,j}^{(t)}, \tag{12}$$

where $w_{m,i}^* = \exp(w_{m,i})/\sum_{i'} \exp(w_{m,i'})$, and $w_{m,i} = \mathbf{w}_m[i]$. Note that $w_{m,i}^*$ is interpreted as a probability that action rule $C_i \in \mathcal{C}$ is the $m$-th component of the policy. Now we complete the 1-step forward reasoning by combining the results from different weights:

$$r_j^{(t)} = softor^\gamma(h_{j,1}^{(t)}, \dots, h_{j,M}^{(t)}). \tag{13}$$

Taking $softor^\gamma$ means that we compose the policy using $M$ softly chosen action rules out of $C$ candidates of rules.

**[(iii) Multi-step reasoning]** We perform $T$-step forward reasoning by computing $r_j^{(t)}$ recursively for $T$ times: $v_j^{(t+1)} = softor^\gamma(r_j^{(t)}, v_j^{(t)})$. Finally, we compute $\mathbf{v}^{(T)} \in [0,1]^G$ and returns $\mathbf{v}_A \in [0,1]^{G_A}$ by extracting only output for action atoms from $\mathbf{v}^{(T)}$. The whole reasoning computation Eq. 8-13 can be implemented using only efficient tensor operations. See App. E.2 for a detailed description.

## E.2 Implementation Details

Here we provide implementational details of the differentiable forward reasoning. The whole reasoning computations in NUDGE can be implemented as a neural network that performs forward reasoning and can efficiently process a batch of examples in parallel on GPUs, which is a non-trivial function of logical reasoners.

Each clause $C_i \in \mathcal{C}$ is compiled into a differentiable function that performs forward reasoning using the tensor. The clause function is computed as:

$$\mathbf{C}_i^{(t)} = softor_3^\gamma \Big( prod_2 \big( gather_1(\tilde{\mathbf{V}}^{(t)}, \tilde{\mathbf{I}}) \big) \Big), \tag{14}$$

where $gather_1(\mathbf{X}, \mathbf{Y})_{i,j,k,l} = \mathbf{X}_{i,\mathbf{Y}_{i,j,k,l},k,l}$ [4] obtains valuations for body atoms of the clause $C_i$ from the valuation tensor and the index tensor. $prod_2$ returns the product along dimension 2, *i.e.* the product of valuations of body atoms for each grounding of $C_i$. The $softor^\gamma$ function is applied along dimension 3, on all the grounding (or possible substitutions) of $C_i$.

$softor_d^\gamma$ is a function for taking logical *or* softly along dimension $d$:

$$softor_d^\gamma(\mathbf{X}) = \gamma \log \big( sum_d \exp(\mathbf{X}/\gamma) \big), \tag{15}$$

where $\gamma > 0$ is a smoothing parameter, $sum_d$ is the sum function along dimension $d$. The results from each clause $\mathbf{C}_i^t \in \mathbb{R}^{B \times G}$ is stacked into tensor $\mathbf{C}^{(t)} \in \mathbb{R}^{C \times B \times G}$.

Finally, the $T$-time step inference is computed by amalgamating the inference results recursively. We take the softmax of the clause weights, $\mathbf{W} \in \mathbb{R}^{M \times C}$, and softly choose $M$ clauses out of $C$ clauses to compose the logic program:

$$\mathbf{W}^* = softmax_1(\mathbf{W}). \tag{16}$$

where $softmax_1$ is a softmax function over dimension 1. The clause weights $\mathbf{W}^* \in \mathbb{R}^{M \times C}$ and the output of the clause function $\mathbf{C}^{(t)} \in \mathbb{R}^{C \times B \times G}$ are expanded (via copy) to the same shape $\tilde{\mathbf{W}}^*, \tilde{\mathbf{C}}^{(t)} \in \mathbb{R}^{M \times C \times B \times G}$. The tensor $\mathbf{H}^{(t)} \in \mathbb{R}^{M \times B \times G}$ is computes as

$$\mathbf{H}^{(t)} = sum_1(\tilde{\mathbf{W}}^* \odot \tilde{\mathbf{C}}), \tag{17}$$

where $\odot$ is element-wise multiplication. Each value $\mathbf{H}_{i,j,k}^{(t)}$ represents the weight of $k$-th ground atom using $i$-th clause weights for the $j$-th example in the batch. Finally, we compute tensor $\mathbf{R}^{(t)} \in \mathbb{R}^{B \times G}$ corresponding to the fact that logic program is a set of clauses:

$$\mathbf{R}^{(t)} = softor_0^\gamma(\mathbf{H}^{(t)}). \tag{18}$$

With $r$ the 1-step forward-chaining reasoning function:

$$r(\mathbf{V}^{(t)}; \mathbf{I}, \mathbf{W}) = \mathbf{R}^{(t)}, \tag{19}$$

we compute the $T$-step reasoning using:

$$\mathbf{V}^{(t+1)} = softor_1^\gamma \Big( stack_1 \big( \mathbf{V}^{(t)}, r(\mathbf{V}^{(t)}; \mathbf{I}, \mathbf{W}) \big) \Big), \tag{20}$$

where $\mathbf{I} \in \mathbb{N}^{C \times G \times S \times L}$ is a precomputed index tensor, and $\mathbf{W} \in \mathbb{R}^{M \times C}$ is clause weights. After $T$-step reasoning, the probabilities over action atoms $\mathcal{G}_A$ are extracted from $\mathbf{V}^{(T)}$ as $\mathbf{V}_A \in [0,1]^{B \times G_A}$.

## F Figures for Ablation Study

Here we provide figures we used for our ablation study.

---

[4] done with pytorch.org/docs/torch.gather

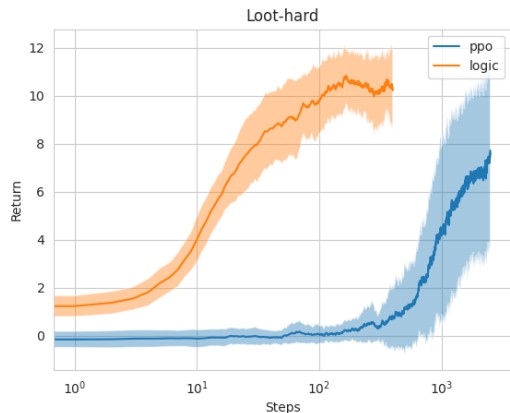

Figure 10: Returns (avg.±std.) obtained by NUDGE and neural PPO on Loot-hard.

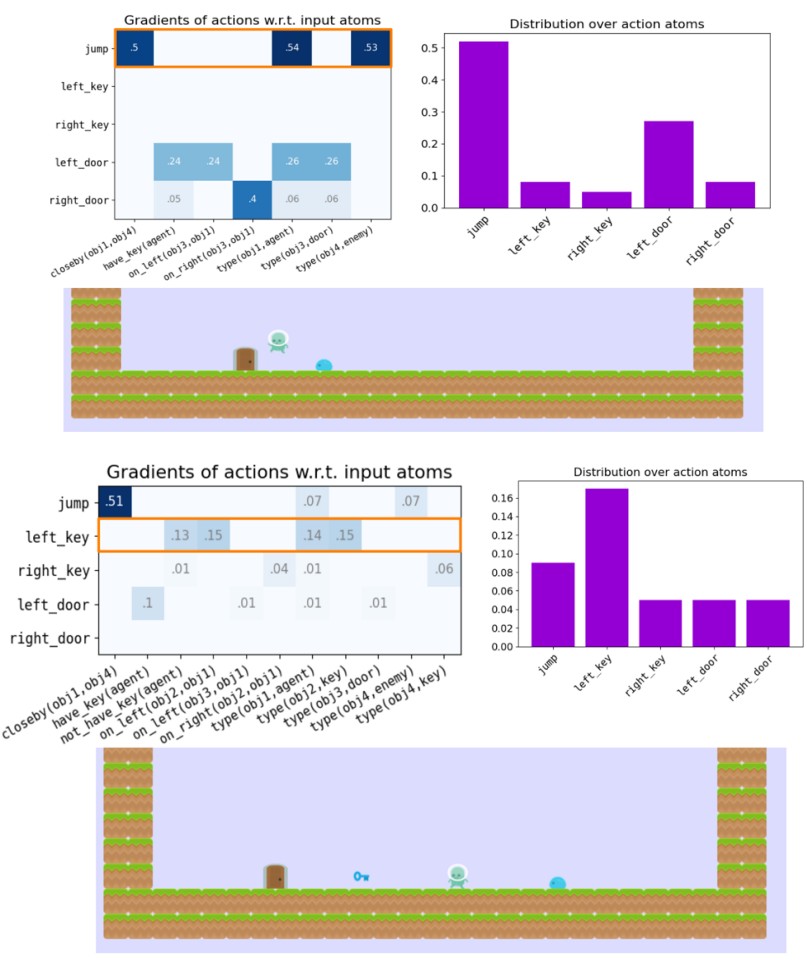

Figure 11: Explanations and action distributions produced by NUDGE for 2 different states of GetOut.

