# OpenReview forum: "Interpretable and Explainable Logical Policies via Neurally Guided Symbolic Abstraction"
_NeurIPS.cc/2023/Conference — NeurIPS 2023 poster_

### Official Review · Reviewer_hGCz · 2023-06-29

**Soundness:** 2 fair
**Presentation:** 3 good
**Contribution:** 3 good
**Rating:** 7
**Confidence:** 4

**Summary:**

The paper tackles interpretable RL with a neurosymbolic method (NUDGE) separating perception and action prediction. Given a pre-trained actor-critic model and a pre-trained/given perception module, NUDGE checks randomly generated rules and assess their quality by comparing them to the pre-trained actor. It then performs rule weight learning using an A2C-like algorithm using the pre-trained critic.

**Strengths:**

I like the ideas, and the paper offers an excellent combination of methods with clear benefits in terms of explainability and possibly generalisation (although I think the verdict is out on this). The paper is well-written, easy to understand and complete.
The explicit parameter $k$ allows limiting the complexity of policies, generating interpretable policies as highlighted in Figure 4.

**Weaknesses:**

The experimental setup varies significantly throughout the different settings, leaving me with questions about the experimental methodology.

**Questions:**

- It is not immediately clear what the differences between the Differentiable Forward Reasoning module are with AlphaILP.
- The object-centric states are similar to concept-based models (for instance, [2], which also performs rule learning). It might be worth discussing.
- 3.2.1: NUDGE never removes state atoms?
- 3.2.1: If $\pi_\theta$ rarely picks action $A$ in states $X$, is the eval rule still reliable?
- Experiments:
	- If all networks get object-centric descriptions, how are those in $[0, 1]$? Or are all these binary?
	- What is the actor-critic/PPO model you use to train NUDGE?
	- What is the motivation behind adding basic rules? It might be worth highlighting this is an option to improve performance or mention that it does not matter.
	- Table 1 Left: Why not compare to PPO? Why did you use expert supervision here? That means you did not test your actual method.
	- Figure 3: The comparison by steps is somewhat unfair because NUDGE uses a pretrained model at initialisation. Why not train all models until convergence?
	- The conclusion that NUDGE adapts to environmental changes is misleading: Rather, it is easy to adapt by changing the logic, but NUDGE is not adaptive itself.
	- The explainability claims are only tested with a single example. It is unclear why NUDGE would be more explainable with LIME-style diagnostics than a neural network trained on object-centric states.
- Table 2 claims NUDGE is FOL, but it is weaker: It does not support infinite domains or function symbols

Minor comments:
- Make sure to use \citep consistently
- The statement that humans use symbolic languages to compose their reasoning is something I'd like to see a citation for
- Throughout the paper, you refer to the rules and their weights as 'probabilistic'. However, $\alpha$ILP is not a probabilistic logic (in fact, the softOr is not fuzzy either, as it is not a t-conorm [3]). The invalid use of the word probability is confusing.
- Typos:
	- 123 reads wrong to me
	- 188: a given neural policy
	- Equation 3 seems invalid. What does the = mean between the $\prod$ and $\mathbf{v}$?
	- Figure 3: Is the largest $k$ 5 or 10? The legend and caption don't agree.
	- 245: designed to
	- 255: Agents can 'open' keys?
	- 298: opening quotes

[1]: Hikaru Shindo, Viktor Pfanschilling, Devendra Singh Dhami, Kristian Kersting: alphaILP: Thinking Visual Scenes as Differentiable Logic Programs.  Machine Learning (Special Issue on Learning and Reasoning 2022), vol. 112, pp. 1465–1497, 2023

[2] Barbiero, P., Ciravegna, G., Giannini, F., Zarlenga, M. E., Magister, L. C., Tonda, A., ... & Marra, G. (2023). Interpretable Neural-Symbolic Concept Reasoning. arXiv preprint arXiv:2304.14068.

[3] van Krieken, E., Acar, E., & van Harmelen, F. (2022). Analyzing differentiable fuzzy logic operators. Artificial Intelligence, 302, 103602.

**Limitations:**

The method requires a (pretrained / hand-designed) perception module that can be hard to come by. The paper does not discuss this limitation thoroughly. Otherwise, limitations are addressed.

---

> ### Author Rebuttal · Authors · 2023-08-09
>
> We thank the reviewer for the detailed feedback and for acknowledging that the paper is well-written, clear, and complete.
> We would like to answer questions and concerns raised in the review.
>
> ***How is the reasoning module different from alphaILP’s one?***
>
>
> NUDGE focuses on differentiable policy learning, i.e. we modified the reasoning module to produce a valid action distribution. Differentiable forward reasoners, in general, output all consequent atoms, but NUDGE outputs only action atoms to achieve a valid action distribution, differentiating action atoms and state atoms.
>
> The object-centric states are similar to concept-based models (for instance, [2], which also performs rule learning). It might be worth discussing.
> This is true, thank you. We are adding the reference and following discussion to the manuscript:
> Concept-based models aim to make DNNs more interpretable by encoding human-understandable concepts into the neural network. NUDGE could rely on concept-based models to extract (weighted) logical representations from raw inputs, e.g., images.
>
> ***NUDGE never removes state atoms?***
>
>
> You are right, NUDGE never removes state atoms. Adding such a capability is easily doable and would be very interesting in situations for which state atoms are not needed. It would simplify the interpretations of the decision-making process.
>
> ***If the neural policy rarely picks action A in states X, is the eval rule still reliable?***
>
>
> Good question. If action A is rarely picked in states X, rules defining action A will get low scores, as the score is based on the action distribution. Every action thus gets a score for every state X that is used in the rule evaluation.
>
>
> ***If all networks get object-centric descriptions, how are those in [0,1]? Or are all these binary?***
>
>
> We add (small) noise to the object-centric description of NUDGE input system, their values thus lie in ]0, 1[.
>
>
> ***What is the actor-critic/PPO model you use to train NUDGE?***
>
>
> Contrary to the neural model that uses an MLP both for actor and critic, NUDGE agents encompass αILP (a differentiable rule learning system) -based actor. Both agent types follow the PPO algorithm (i.e., use clipping to stabilize policy optimization).
>
> ***What is the motivation behind adding basic rules? Does it improve performance?***
>
>
> Excellent question. NUDGE can start from the set of empty rules for each action.
> We provided the first rule-mining step in our manuscript as it helps to understand how the rules are expanded.
>
> ***Tab. 1 Left: Why not compare to PPO? Why did you use expert supervision here?***
>
>
> We used DQN, as it is the most used baseline in the literature for this benchmark. The PPO results reported in its introductory paper ([1]) do not change our conclusions.
> Atari's games are a common RL benchmark. Compared to our implemented environments, they rely less on relational and logic reasoning, but require accurate moves. Our Atari experiments are meant to show that αILP agents can learn on this benchmark too. We have observed that NUDGE produces equivalent rules as the one provided by the game experts in all other tested environments, explaining why we have not run the rule search yet.
> For completeness, we are currently setting up and running these experiments to be added to the final version.
>
>
> [1] Schulman, J., et al.: Proximal policy optimization algorithms. (2017)
>
> ***Fig. 3: Why not train all models until convergence?***
>
>
> The reported results in the tables are from agents trained until convergence. NUDGE does not use a pretrained model, starts with a randomly initialized critic, too. However, its actor is composed of already curated rules, as “NUDGE exploits trained neural network-based agents.” (l. 7)
> Agents that use expert priors do not make use of this advantage but incorporate expert knowledge. We show the learning through steps allows the reader to see how much prior knowledge can help to boost learning.
>
> ***The conclusion that NUDGE adapts to environmental changes is misleading: Rather, it is easy to adapt by changing the logic, but NUDGE is not adaptive itself.***
>
>
> Very important insight. It depends on the environment.
> In our experiments, in the GetOut environment, the agent that is trained using 1 moving enemy can perform well in the variant with more moving and static enemies. This is because of the NUDGE’s symbolic abstraction and concepts, e.g., closeby, are good enough to adapt to more enemies. The modified Loot environments did not need any change either.
> Indeed, in 3Fishes-C, the agent requires a change in its logic to adapt. This change was made manually, but the new one could be obtained by deriving from the existing one. It would be faster than fully retraining an agent. We will clarify this in the paper.
>
>
> ***Why would NUDGE be more explainable than neural networks with LIME?***
>
>
> Due to the space limitation, we put only one example in the manuscript.
> However, we conducted the experiments for different actions producing different explanations. In Fig.2 in the attached PDF, we show 2 more explanations and action distributions produced by NUDGE. For each state, NUDGE produced human-readable explanations using gradients and logical representations, which are difficult to be produced by neural networks. Thank you for pointing it out. We will add these results to the appendix.
>
> ***Tab 2 claims NUDGE is FOL, but it is weaker: It does not support infinite domains or function symbols***
>
>
> Good point, thank you for raising this. Currently, we are not providing experiments using functors or infinite domains. NUDGE can handle these as it incorporates the differentiable ILP techniques for functors [1], where necessary symbols are adaptively enumerated according to input.
> We will add this discussion to the manuscript.
>
> [1] H. Shindo, et al.: Differentiable Inductive Logic Programming for Structured Examples AAAI 2021
>
> Finally, we greatly appreciate all the minor comments. We address them in the updated version.

---

> > ### Comment · Reviewer_hGCz · 2023-08-17
> >
> > I thank the authors for their detailed response and the dedication to implementing some suggestions in the final paper.
> >
> > > Good question. If action A is rarely picked in states X, rules defining action A will get low scores, as the score is based on the action distribution. Every action thus gets a score for every state X that is used in the rule evaluation.
> >
> > Could this incur an exploration issue?
> >
> > > The PPO results reported in its introductory paper ([1]) do not change our conclusions
> >
> > Why not add it to the results for completeness, then?
> >
> > > For completeness, we are currently setting up and running these experiments to be added to the final version.
> >
> > That sounds good!
> >
> > > For each state, NUDGE produced human-readable explanations using gradients and logical representations, which are difficult to be produced by neural networks
> >
> > These explanations are indeed useful, but I do not understand why neural networks could not provide these explanations. The neural network may certainly implement a policy that is behaviourally similar to NUDGE. Then its gradients must also be similar and hence the LIME explanations will be.
> >
> > To comment on the discussion of lack of experiments on unstructured inputs: In neuro-symbolic systems, many papers focus on end-to-end training of the perception and the logical modules. I agree with the authors that there are also many systems that fix pre-trained perception modules, like NUDGE. The presented method is itself neurosymbolic (it combines differentiable/neural rule learning with explicit symbolic rules). In the vision that pre-trained neural perception is enough, the presented method makes sense, although I agree the writing could make this vision clearer.

---

> > > ### Author Response · Authors · 2023-08-19
> > >
> > > We thank the reviewer for the feedback and the support for the fact that NUDGE is a valid neuro-symbolic system. We now answer your new points.
> > >
> > > ***Could this incur an exploration issue?***
> > >
> > > Very important question, thank you for asking this.
> > > In our experiments, we performed the neurally-guided symbolic abstraction for each different action, respectively.
> > >
> > > For example, in the GetOut environment, we generate candidate rules for action jump, then obtain, e.g., 50 rules. After that, we generate rules for action right, etc.
> > > This can be done efficiently since NUDGE uses the cached history of states and actions produced by the neural policy.
> > > Even if action right appears more often than jump, each rule will be evaluated accordingly. More cached history for action $A$ will improve the search quality for rules defining action $A$.
> > >
> > > If the cached history does not contain any record of action $A$, then the candidate rules about action $A$ would get almost the same scores, leading to a random search (if 2 rules have the same score, then 1 of them is randomly selected to be expanded.).
> > > We added this assumption to the new version of our manuscript.
> > > The question helped us a lot to clarify the assumption of our method. Thank you.
> > >
> > > ***Why not add the PPO results for completeness?***
> > >
> > > You are right, we added them for completeness to our manuscript.
> > >
> > > ***Why neural networks could not provide these explanations?***
> > >
> > > Good point. In our experiments, we observed that the neural agent produced noisy explanations. Let us demonstrate.
> > >
> > > We computed explanations of a neural ppo agent on a state of GetOut depicted in Fig.1, and the correct explanation is to go right *to get the key, and the key is located on the right of the agent*. We use the input-gradients method for a fair comparison since it is used for NUDGE explanations. These are local gradients and thus can be considered similar to LIME based explanations.
> > >
> > > We show the concepts which received high scores by the explanation with their scores :
> > > - (1.22) key x-axis position **[correct]**
> > > - (1.21) key x-axis velocity
> > > - (1.10) enemy y-axis velocity
> > > - (0.77) door x_axis position
> > > - (0.66) door y_axis position
> > > - (0.56) door y_axis velocity
> > > - (0.47) agent x-axis position **[correct]**
> > > - (0.34) key y-axis velocity
> > >
> > > (We note that neural agents take also the velocity of each object.)
> > >
> > >
> > > It correctly highlights the concept of *key x-axis position*. However, the necessary concepts, such as the *agent x-axis position*, are evaluated lower than other irrelevant concepts.
> > >
> > > Moreover, the explanation contains a lot of noise, e.g.,  *key x-axis velocity* (which is always 0) is highlighted, and the position and velocity of the door are also highlighted. This shows that neural agents failed to produce clear explanations, wrongly focusing on unimportant concepts.
> > >
> > > It is known that neural networks trained on logical concepts can produce wrong explanations often. The paper [1] shows exactly this behavior, claiming that additional regularization/supervision about explanations during training is necessary to force neural networks to produce correct explanations. This approach requires additional efforts to label each state with its correct explanation.
> > >
> > > We added the result and discussion to our manuscript. Thank you for indicating points that are very helpful in improving the manuscript.
> > >
> > >
> > > [1] Wolfgang Stammer, Patrick Schramowski, Kristian Kersting: Right for the Right Concept: Revising Neuro-Symbolic Concepts by Interacting With Their Explanations. CVPR 2021

---

> > > > ### Comment · Reviewer_hGCz · 2023-08-19
> > > >
> > > > I thank the authors for engaging thoroughly with the rebuttal and addressing my concerns. The ideas about exploration and the explanations of neural agents are useful additions to the paper.
> > > >
> > > > The only concern I have left is if the authors could clarify that this paper assumes a pretrained perception module. I already shared my opinion on that I believe this assumption is completely fine, but (as I mentioned in the limitations section of my original review) this is not clear enough in the paper. In my opinion, this should be part of the abstract: It is vital information for whether someone would want to dive into this paper. The abstract mentions "especially when working on the image level", slightly suggesting otherwise.

---

> > > > > ### Author Response · Authors · 2023-08-19
> > > > > **Thank you for the comment. We will clarify in abstract**
> > > > >
> > > > > We would like to thank the reviewer for their encouraging words and finding that we have resolved their concerns.
> > > > >
> > > > > Yes we will clarify this in the abstract and explicitly state that we use a pre-trained perception module in the final version of the paper. Apologies for missing this in your initial review. We had mentioned this in the 1st point of our contribution list (l.54-55): **NUDGE uses neurally-guided symbolic abstraction to efficiently find a promising ruleset using pretrained neural-based agents guidance.** We will also make this clearer.
> > > > >
> > > > > We hope that we have now clarified all of your concerns and you reconsider your score. It was a pleasure having a discussion with you and all your comments have already helped making the manuscript clearer.

---

> > > > > > ### Comment · Reviewer_hGCz · 2023-08-21
> > > > > >
> > > > > > I thank the authors, indeed it was a good discussion! The authors have addressed my main concerns and I will raise my score to 7 and increase confidence to 4.

---

> > > > > > > ### Author Response · Authors · 2023-08-22
> > > > > > > **Thank you again**
> > > > > > >
> > > > > > > We would like to thank the reviewer for the increase in the score and most importantly for the great interaction.

---

### Official Review · Reviewer_Jkdb · 2023-06-29

**Soundness:** 4 excellent
**Presentation:** 3 good
**Contribution:** 3 good
**Rating:** 6
**Confidence:** 3

**Summary:**

This paper presents NUDGE, an RL technique that combines neural network and logic program learning to learn policies that are interpretable and explainable by humans. NUDGE produces agents that match or exceed the performance of agents trained using standard RL techniques like PPO.

**Strengths:**

The paper is well-written, clear, and addresses the significant challenge of uninterpretable reinforcement learning policies. The paper proposes to solve this by learning policies that are interpretable and explainable by design.

I am unqualified to assess the originality of this paper because I am unfamiliar with the logic program learning literature.

**Weaknesses:**

This work can be improved with stronger definitions of interpretability and explainability. Currently, it’s unclear what criteria make a model interpretable or uninterpretable (line 43). What exactly makes a policy readable by humans? Is this person an expert or a layperson? I suspect that interpretability here means that an expert can assign meanings to the policy. Similarly, what constitutes an explanation (line 44)? From their definition, it seems that the explanation need not be causal? Do all methods from explainable AI count?

There are also existing definitions of both terms from the community. I would like to see the authors engage with the existing literature on definitions of interpretability and explainability. See Lipton 2016 for a discussion on interpretability. I think that having set definitions is important for this work in order to avoid moving goalposts.

Line 141: Another weakness is that the relational perception module maps all entities to a particular index in the valuation vector. This method does not seem like it would scale to new objects. Similarly, the identity of entities seem to be fixed given a particular validation vector. It seems like the parallel version of this weakness (parallel in the sense of scaling issues) for the forward reasoning module is solved by neurally guided symbolic abstraction (line 193).

I also found some parts on what parameters are and aren't learned to be confusing. See questions.

**Questions:**

- Line 3: What do you mean by “articulate the thinking” in this definition?
- How do you envision this work translating to real world settings? This work makes a lot of sense in the rule-based testbed, where much of the non-rule-based information is extraneous. Can naturalistic inputs be represented in this way? I could see an argument for this work as automating the pipeline approaches in vision-based control settings.
- Line 142: I’m curious whether there are analogues in cognitive science for these modules you’ve created.
- Line 143: Is $\Theta$ learned?
- Line 204: What is $\theta$? I don’t see $\theta$ anywhere else.

Nits:
- Line 1: “dominating” —> “dominant?

**Limitations:**

The authors have adequately addressed limitations.

---

> ### Author Rebuttal · Authors · 2023-08-09
>
> We thank the reviewer for the detailed feedback and for finding that the paper is well-written, clear, and addresses the significant challenge of uninterpretable reinforcement learning policies.
>
> We address concerns next.
>
>
> ***Engaging with the existing literature on definitions of interpretability and explainability***
>
>
> Thank you for this resource. Explainability and interpretability are indeed used interchangeably in the literature. We thus set their definition (following [1]):
> * **interpretability**, i.e. the capacity to articulate the thinking behind the action selection. (l.16)
> * **explainability**, i.e. [the ability of] the agent can explain the importance of each input on its decision. (l.33)
> These definitions align with the ones discussed by Lipton (2017). We engage more research works (including the provided one), to improve our discussion on these terms for the final version.
>
> [1] Rudin, C. (2019). Stop explaining black box machine learning models for high stakes decisions and use interpretable models instead. Nature machine intelligence, 1(5), 206-215.
>
>
> ***Relational perception module maps all entities to a particular index in the valuation vector.***
>
>
> Indeed, to perform reasoning, we need to name entities and relations. It would be very interesting to investigate how we can remove mappings by incorporating deep representations obtained by, e.g., object-discovery models. We left it for future work. Thank you for the insight.
>
>
> ***Some parts on parameter learning confusing***
>
> We appreciate your detailed feedback, we clarify hereafter and did improve the paper based on your questions:
> * **What do you mean by “articulate the thinking” in this definition? (l. 3)**
>    -> explain the reasoning behind [the action selection]
> * **I’m curious whether there are analogues in cognitive science for these modules you’ve created (l. 142).**
> This is a very exciting question, we have found [1] and [2] and more recently [3] (also more focus or AI) that provides some light on this matter. It seems that such module are simplifications of what is happening inside biological nervous systems. The work of Daniel Kahneman is also of relevance here.
> * **Is $\Theta$ learned ? (l. 143)**
> $\Theta$ is a parameter for the perception networks (e.g. an object detector), and we assumed that it is pre-trained on the environment. We will clarify this more in the manuscript.
>  * **What is $\theta$, I don’t see it anywhere else. (l. 204)**
> $\theta$ represents a parameter of neural policy $\pi_\theta$, i.e. the set of learnable weights in the neural agent. We assume $\theta$ is pre-trained, and NUDGE produces differentiable logic policy $\pi_{(\mathcal{C}, \mathbf{W})}$, which is parameterized action rules $\mathcal{C}$ and their weights $\mathbf{W}$. It is introduced l. 194.
>  * **How do you envision this work translating to real world settings? This work makes a lot of sense in the rule-based testbed, where much of the non-rule-based information is extraneous. Can naturalistic inputs be represented in this way? I could see an argument for this work as automating the pipeline approaches in vision-based control settings.**
> Our work rely on an accurate object-centric and/or concept extraction method to produce successful and explainable policies. For general real world agents, this would be tedious, but techniques to ease the extraction of concepts based on, e.g., natural language are already emerging. You are right that the automatic pipeline approach in industrial, based on, e.g., vision sensors is a very interesting real world use case that would fit NUDGE policies’ advantages.
>
> [1] Alvarez, M. (2018). Reasons for action, acting for reasons, and rationality. Synthese, 195, 3293-3310.
>
>
> [2] Ryan, R. M., & Connell, J. P. (1989). Perceived locus of causality and internalization: examining reasons for acting in two domains. Journal of personality and social psychology, 57(5), 749.
>
>
> [3] Huang, J., Zhu, W. Y., Jia, B., Wang, Z., Ma, X., Li, Q., & Huang, S. (2022). Perceive, Ground, Reason, and Act: A Benchmark for General-purpose Visual Representation.

---

> > ### Comment · Reviewer_Jkdb · 2023-08-13
> > **Keeping my score the same**
> >
> > Thank you to the authors for adequately addressing my concerns. I defer to the other reviewers in assessing the novelty/originality of this work and keep my score the same.

---

### Official Review · Reviewer_AhLx · 2023-07-03

**Soundness:** 2 fair
**Presentation:** 1 poor
**Contribution:** 2 fair
**Rating:** 2
**Confidence:** 4

**Summary:**

This paper aims to learn more interpretable and explainable logical policies. It modifies NLRL, Neural Logical RL, with (1) an extra neural-guided heuristic, i.e., alignments with a pretrained neural policy, for filtering logical rules during search; (2) changing policy learning algorithm to actor-critic from policy gradients; (3) pre-defining / hardcoding the meanings of symbols for more interpretability; and (4) explanation of policies according to the gradients. It conducted experiments in five simple games.

**Strengths:**

This paper uses pre-trained policy networks as neural guidance for faster search of logical rules (filtering according to their alignments). This could be an interesting direction to explore. However, experiments that evaluate/support its effectiveness are not adequate in the paper.

**Weaknesses:**

This paper claims that their methods improve the interpretability and explainability of logical policy learning. However, as far as I understand, this is not true because
* The proposed methods (compared with existing ones) are mainly about optimization and logical rule mining, instead of about interpretability and explainability.
* Policies become more interpretable only because they change the problem setting to an easier and more interpretable one. They assume predefined/hardcoded meanings of all the predicates/symbols (e.g., `close_by(obj1, obj2)`), as well as a pretrained neural module to detect the objects and those meaningful attributes.
  - Previous methods do not assume those and instead, jointly learn the symbols and their meanings with logical rules.
  - Any model with access to those predefined meanings and those pretrained detection modules is more interpretable and explainable. For example, when given access to those symbolic inputs (i.e., memory states), even DQN can adapt to environmental changes (as defined in Q2, Sec. 4) and would be explainable by performing the gradient analysis.

From the optimization perspective, the new problem setting is less interesting and challenging as well. With access to symbolic inputs (e.g., the memory states of Atari games), classic ILP methods may even work, especially when considering the relatively small search space in the five simple games. The authors should compare with those baselines as well.

There are other minor concerns such as
* If the neural policies are much worse than the symbolic ones as reported, why and how would it be helpful to distill/use them as neural guidance?
* The games are relatively simple and easy.
* There should be experiments analyzing the effects of actor critics v.s. policy gradients.


**Questions:**

* Why are the policies more interpretable and explainable? How can the proposed methods contribute to them?
* Are there results of classic symbolic methods? Can NUDGE perform better than them?
* Are there results on more complex and difficult games?

---

> ### Author Rebuttal · Authors · 2023-08-09
>
> We thank the reviewer for the detailed feedback and for highlighting that our proposed approach is interesting.
> We address the concerns next.
>
> ***Proposed methods not about interpretability and explainability.***
>
> One of our main contributions is indeed the neural guidance incorporated in our NUDGE framework. This is why the neural guidance is part of the name of our framework, whereas Interpretability and Explainability are not.
>
> However, our motivation is to develop interpretable and explainable policies for RL, and the optimization is a means to an end.
> Table 2 shows that NUDGE, is the first interpretable and explainable RL framework using differentiable logic reasoning. We demonstrate empirical support for these features (Fig. 4 for interpretability, Fig. 5 for explainability).
> The system's methodology and outcome are orthogonal, i.e., we perform differentiable rule learning to obtain policies that address the interpretability and explainability problems.
>
> ***Changing the problem setting to easier and more interpretable.***
>
> Adapting the problem settings by focusing on object-centric representation is a common way of obtaining interpretability.
> As explained in the paper, we believe that obtaining an interpretation over the RL agents’ behaviors requires this problem-setting change. We are not hiding this point, as stated l. 147, “NUDGE agents take an object-centric state representation as input, obtained e.g. using object detection or discovery methods.” We have not incorporated such techniques yet.
>
> For a fair comparison, we used object-centric input representations for neural policies. However, structured inputs are not sufficient to obtain interpretability, as the trained neural baselines are not interpretable (due to their black-box nature), whereas NUDGE policies are.
>
> The use of predefined (and meaningful) symbols is a fundamental feature, and often a drawback, of any logic-based system.
> As mentioned in the limitation, predicate invention could be an alternative that would lower the priors provided to logic-based agents. We left this for future work.
>
> Finally, NUDGE has an explicit logical reasoning structure and thus can produce the reasoning steps of action selection, explaining the reasoning behind decision-making. Building precise reasoning steps using neural networks is still an open problem.
>
> ***Classic ILP methods may work + comparison to these***
>
> Learning in RL tasks using symbolic ILP solvers is very inefficient, as their non-differentiable nature prevents the use of efficient RL algorithms (such as policy-gradients techniques). One needs to construct inputs that can be processed by ILP solvers i.e., positive/negative labeled examples, namely transforming a RL task to a binary classification task.
>
> Moreover, even with a provided set of action rules obtained from trained NUDGE agents, the classic ILP framework’s deterministic reasoning will not allow for rule prioritization.
> For example, the weighted rules of NUDGE policies in Getout are:
>
> * 0.57:jump():-...
> * 0.29:right_key():-...
>
> These weights show that the agent has learned to prioritize jump over right_key. These adaptive action selections cannot be done easily by classic ILP.
>
>
> To demonstrate this, we conducted reasoning experiments using trained NUDGE agents and Classic logic benchmark, where the set of action rules are given but with all weights set to 1.0, which simulates discrete logic. Note that such agents can still play better than ones with pure symbolic reasoners, as our classic baselines still use fuzzy (i.e., continuous) concepts, such as closeby.
> We show the average (and std) returns of 5 randomly seeded agents on GetOut, 3Fish, and Loot. We used action rules obtained by NUDGE.
>
> | Score | NUDGE          | Classic        | Random         |
> |-|-|-|-|
> | GetOut         | 17.86±2.86 | 11.59±4.29 | $-22.5±0.41 |
> | 3Fish          | 0.05±0.27 | -0.24±0.29 | $-0.73±0.05 |
> | Loot           | 5.66±0.59  | 0.51±0.74  | $0.57±0.39  |
>
> In all environments, NUDGE clearly outperforms the Classic counterparts.
> This experiment already shows the superiority of differentiable policy reasoning (embedding weighted action rules) over policy without weighted rules (Classic).
>
> Finally, learning using classic ILP relies on policy reasoning, and thus will lead to (at best) brittle learning.
>
> As shown above and explained in [1], it is neither trivial nor efficient to use classic ILP solvers in such RL tasks, we do not see why we should develop and compare with these methods. Adding these results and discussions will improve the manuscript.
>
> [1] Zhengyao Jiang and Shan Luo: Neural logic reinforcement learning. ICML 2019
>
> ***Why and how would it be helpful to distill/use neural policies as guidance***
>
>
> Neural policies are used to find the set of promising rules that NUDGE agents will use when learning to solve the task.  Of course, the better the neural agents are, the more likely is the set of rule to be successful.
> NUDGE agents then have the advantage of relying on logic rules, and their embedded priors, which allows for faster and better generalization.
>
> ***The games are relatively simple and easy.***
>
>
> We have added experiments on a harder version of loot (loot-hard, *see answer to reviewer AnR8 or the attached PDF for details*).
> The set of games used in our experiments are meant to back up our claims, i.e., demonstrate that we can produce explainable and interpretable logic based policies that solve RL tasks that involve relational reasoning. Scaling it to more complex games (such as starcraft) is left for future work.
>
> ***Experiments analyzing the effects of actor critics v.s. policy gradients.***
>
>
> We directly incorporated the actor-critic framework because we use the critic in the neurally guided symbolic abstraction. The critic stabilizes the gradients backpropagated to the policy compared to direct backpropagation from the return. The focus of our paper is not to compare the advantages of existing RL methods.

---

> > ### Comment · Reviewer_AhLx · 2023-08-13
> > **Reduce the rating to 2 (strong reject)**
> >
> > Most importantly, **Symbolic policies and Neuro-Symbolic policies are totally different.**
> > * In short, "symbolic v.s. neuro-symbolic policies" is similar to "linear regression v.s. neural-network-based policies."
> > * Symbolic policies receive structured inputs, e.g., the predefined object-centric representations, while neuro-symbolic policies receive unstructured inputs such as high-dimensional unstructured images.
> > * Neuro-symbolic policies are much more challenging and practically difficult to learn than symbolic ones because of (1) the curse of dimensionality with high-dimensional inputs, (2) the ill-posed problems for discovering meaningful latent representations, (3) searching policies with both discrete (e.g., logic structure) and continuous (e.g., neural network weights) parameters, and many others.
> > * Neuro-symbolic policies are also much more practical and meaningful than pure symbolic ones. They are basically using the same experimental setting as Deep RL, while trying to obtain better interpretability, explainability, generalizability, transferability, and so on (instead of "sacrificing" the experimental settings to achieve them, as the pure symbolic policies). In many real environments, structured inputs (as those semantically labeled predicates) are unavailable or hard to define/supervise. It is thus preferable to learn them using neuro-symbolic methods. The same reason for us to prefer learning neural features over predefined ones when achievable.
> >
> > **This paper claims it is and compares itself with Neuro-Symbolic methods. BUT it actually uses a Symbolic policy setting, as stated in a line on page 5 (and as quoted in the rebuttal by the authors.)**
> > * The introduction first starts with Deep RL (talking about unstructured inputs) and then talks a lot about a Neuro-Symbolic method, NLRL. Most of the NLRL's drawbacks as discussed in the paper are about the difficulties to learn a neurosymbolic policy which the proposed method does not solve. Again, the paper avoids those problems by changing to pure symbolic settings from page 5. For example, the drawbacks of NLRL in the introduction include
> >   - line 29: NLRL is a memory-intensive approach,
> >   - line 31: This approach can generate many newly invented predicates without their specification of meaning.
> > * In the introduction, the proposed method, NUDGE, still receives unstructured inputs instead of structured ones.
> >   - line 39: Given an input state, NUDGE extracts entities and their relations, converting raw states to a logic representations. This probabilistic relational states are used to deduce actions...
> > * In Figure 2, the relational perception module, which maps inputs to structured ones, is also illustrated as part of their methods, while this module is actually predefined. The papers seem even assuming semantic labels of all objects/predicates.
> > * In experiments, they just compare with pure neural policies (e.g., DQN) and neuro-symbolic policies (e.g., NLRL).
> >
> > **There are many more advanced methods to learn pure symbolic policies (e.g., [1,2,3]). They are evaluated in more complex domains, e.g., Karel and Karel Hard. This paper does not compare them in its simple environments either.**
> > * For example, LEAPS [1], published in 2021, learned a latent space for symbolic policies and is shown to perform well in Karel.
> >
> > **As a symbolic method, the method itself is neither as interesting nor novel.**
> > * Neural guidance has been explored in symbolic settings [4].
> > * Other advanced symbolic policies have been shown to perform well in more complex environments [1,2].
> > * For clarification, in my review, I was saying "using neural guidance for learning Neuro-Symbolic policies" could be an interesting direction as it is unclear how to use neural guidance while searching both discrete and continuous parameters simultaneously in the neuro-symbolic setting. Using it for learning symbolic policies is not as novel.
> >
> > **Many other concerns of mine (as stated in the review) are not solved but are more likely to be confirmed during the rebuttal period. Therefore, with all these concerns in mind and considering the risk of misleading overclaim presentation, I changed the rating to 2 (Strong Reject).**
> >
> > [1] Trivedi, Dweep, et al. "Learning to synthesize programs as interpretable and generalizable policies." Advances in neural information processing systems 34 (2021): 25146-25163.\
> > [2] Liu, Guan-Ting, et al. "Hierarchical Programmatic Reinforcement Learning via Learning to Compose Programs." arXiv preprint arXiv:2301.12950 (2023).\
> > [3] Qiu, Wenjie, and He Zhu. "Programmatic reinforcement learning without oracles." International Conference on Learning Representations. 2021.\
> > [4] Shah, Ameesh, et al. "Learning differentiable programs with admissible neural heuristics." Advances in neural information processing systems 33 (2020): 4940-4952.

---

> > > ### Author Response · Authors · 2023-08-14
> > > **Complete misunderstanding of our Approach (1/2)**
> > >
> > > We first would like to express our concern about the inconsistency in the response by the reviewer. We have addressed every issue raised in the review, but the reply completely ignores our rebuttal response.  For example, we conducted new experiments based on the reviewer's comments hoping that they would address the concern, but the reviewer posted entirely new comments which do not align with the initial review. Thus it is very challenging for us to rebut convincingly.
> > >
> > > As the reviewer states that several concerns remain, we would like to ask the reviewer to point these out explicitly so that we can have a fruitful and scientific discussion which is the whole purpose of a discussion period. We still address your novel concerns hereafter:
> > >
> > > **This paper claims it is and compares itself with Neuro-Symbolic methods. BUT it actually uses a Symbolic policy setting**
> > >
> > > We disagree and would like to point out that this is a complete misunderstanding of what we do in this work. Differentiable forward reasoning used in NUDGE establishes the **4th type of neuro-symbolic system, i.e., Neuro:Symbolic→Neuro, as proposed by Kautz (2022)**. NUDGE’s policy maps the output of neural networks (Neuro) to symbolic representations (Symbolic), then gradient-based learning is performed on top of it (Neuro).
> > > The weighted logical policy is learned by using gradients, and NUDGE thus constitutes a neuro-symbolic system as differentiable logic is neuro-symbolic. But again, this is not the point of the paper, as NUDGE is a  framework for interpretable and explainable policy reasoning and learning in reinforcement learning as has been correctly pointed out by reviewers NLZb, Jkdb and hGCz.
> > >
> > > **Symbolic policies receive structured inputs while neuro-symbolic policies receive unstructured inputs.**
> > >
> > > As explained in Kautz’s paper, there are 6 different instantiations of neuro-symbolic models. What you have provided is just one instance of them, and unfortunately, inadequate to claim that NUDGE is not neuro-symbolic. There are also other types of neuro-symbolic systems, other than Neuro:Symbolic→Neuro i.e. NUDGE, such as [symbolic Neuro symbolic] where input and output are presented in symbolic form.  **We do not find any references to support the reviewer’s exclusive definition of neuro-symbolic frameworks**. As a counter argument, here is a reference: Neurosymbolic Reinforcement Learning with Formally Verified Exploration, Anderson et al. NeurIPS 2020, that is a **neuro-symbolic** RL method but does not take unstructured input.
> > >
> > > Kautz, H. (2022). The third AI summer: AAAI Robert S. Engelmore memorial lecture. AI Magazine, 43(1), 93–104.
> > >
> > > Most importantly, even if the reviewer’s definition of neuro-symbolic would be the only correct definition, we have **never** stated that our method is neuro-symbolic in our manuscript, but only neurally guided since being neuro-symbolic is not the point of this work. “Neuro-symbolic” appears twice in our paper:
> > > 1. When we state that "neuro-symbolic RL aims at creating policies that are interpretable in the first place" (l. 4)
> > > 2. When we refer to the "NeuroSymbolic RL (NeSyRL)" method of [Kimura et al., 2021]. (l. 336)
> > >
> > > This comment is thus totally irrelevant and does not justify a rejection.
> > >
> > > **NLRL's drawbacks are not solved**
> > >
> > > We disagree with this since NUDGE solves the drawbacks.
> > >
> > > 1. NLRL is a memory-intensive approach. NUDGE uses the efficient neurally-guided symbolic abstraction to manage the search space. Experiments show that NUDGE can learn more complex rules than NLRL. NLRL is evaluated on symbolic abstract environments.
> > > Our statement "line 29: NLRL is a memory-intensive approach" refers to the fact that without neural guidance, every rule has to be generated, as explained directly after: "i.e. it generates a set of potential simple rules based on rule templates" (l. 30).
> > >
> > > 2. NLRL can generate many newly invented predicates without specification of meaning, but NUDGE produces human-readable programs using meaningful predicates. Experiments show that NUDGE learned policies using different predicates for different environments.
> > > We note that NLRL does not learn perception models since they are evaluated on symbolic abstract environments.
> > >
> > > As stated in the introduction, our contribution is the neurally-guided symbolic abstraction using differentiable logic programming, allowing us to learn logic-based policies very efficiently using gradients.
> > >
> > > **In experiments, they just compare with pure neural policies (e.g., DQN) and neuro-symbolic policies (e.g., NLRL).**
> > >
> > > We have also presented results for a purely symbolic baseline in the rebuttal and added it to the new version of the paper but this has been conveniently ignored by the reviewer.

---

> > > > ### Author Response · Authors · 2023-08-14
> > > > **Complete misunderstanding of our Approach (2/2)**
> > > >
> > > > **Other advanced symbolic policies have been shown to perform well in more complex environments [1,2].**
> > > >
> > > > We agree. This shows that our agents could potentially tackle more complex environments if resources were devoted to it.
> > > > Our experimental evaluation is meant to back up our claims but based on the feedback by reviewer AnR8, we have added a more logically challenging environment in the rebuttal with the attached pdf in the global response which is again strangely ignored by the reviewer. We would like the reviewer to point out a paper which tests on all the environments possible or that [1,2] are the standard benchmarks for these papers.
> > > >
> > > > **As a symbolic method, the method itself is neither as interesting nor novel.**
> > > >
> > > > Again NUDGE is a neuro-symbolic method. Moreover, the claim, “Neural guidance has been explored in symbolic settings [4].” is incorrect since the proposed framework, NEAR [4], does not address RL at all and uses functional programs. To the best of our knowledge, NUDGE is the first system that uses Neural Guidance in RL settings using first-order logic with differentiable reasoning.
> > > > Even if parts of NUDGE have (obviously) been studied and developed, our work, in the way it adapts differentiable logic policies based on first-order logic to RL, is novel.
> > > >
> > > > We hope we have presented our arguments clearly and will be happy to clarify further concerns.

---

> > > > > ### Comment · Reviewer_AhLx · 2023-08-14
> > > > >
> > > > > I think I have made my points clear and tried to rephrase the same issues in detail from multiple perspectives so that, hopefully, the authors can understand them and face the issues. I will try the last time to state the issues more explicitly here:
> > > > > * The most important issue is always changing the experimental settings. It claims the method receives unstructured inputs as images but actually uses some pretrained models to obtain the structured ones. As I explained in my last reply, these two settings are totally different: (1) handling low-dimensional structured inputs is much easier than unstructured ones; (2) structured inputs are hard to obtain in many real-world environments, and so on.
> > > > >   - In my original review, this issue is about interpretability and explainability because, if what the paper claimed in the introduction was achieved (i.e., a neuro-symbolic policy, or with unstructured inputs), the model/policy does become much more interpretable & explainable than pure neural ones. Unfortunately, this paper changed the setting to obtain the claimed improvements in interpretability and explainability. It is far from standard to claim those improvements in the introduction "with unstructured inputs" but actually use structured inputs from page 5.
> > > > >   - In my second reply, this issue is phrased as differences between neuro-symbolic and symbolic methods. They have different inputs; one is much more challenging and practical than the other; etc.
> > > > >   - In response to the authors' last reply, regardless of how to define "symbolic" or "neuro-symbolic", the experimental setting is different/changed. It becomes much easier so that the simple environments are not interesting and the method becomes less applicable in practice because the predefined module is not always going to be available or useful.
> > > > > * The other major concern of mine is that, for a method with structured inputs, i.e., symbolic policies, the optimization problem becomes much less challenging and interesting with the current simple environments. Both replies are about them.
> > > > >   - I do see the new environment and results. However, for new environments, I still think it is simple (compared with other environments used with structured inputs). I have nothing else to comment on except for letting AC or other reviewers double-check. For new results, I was surprised that the authors replied with some unknown old methods. $\partial$ILP, published in 2019, is already classic/old from my perspective. There are also recent packages like Scallop [1].
> > > > > * Regarding minor concerns:
> > > > >   - "lame neural policy -> good neural guidance": the authors just replied with some intuitions and perhaps guesses. I cannot verify if they are valid, so the concern still holds.
> > > > >   - "actor critics v.s. policy gradients": the same as before. I have nothing else to say except for receiving some intuitions and perhaps guesses.
> > > > >
> > > > > Anyway, the main issues are always (1) changing the experimental setting to a much easier one and (2) the method is not interesting and the results are not convincing in the actual setting. I hope the authors can see them and should at least change claims about unstructured inputs in the first four pages of the paper, letting the reviewers make decisions upon them.
> > > > >
> > > > > [1] Huang, Jiani, et al. "Scallop: From probabilistic deductive databases to scalable differentiable reasoning." Advances in Neural Information Processing Systems 34 (2021): 25134-25145.

---

> > > > > > ### Author Response · Authors · 2023-08-15
> > > > > > **Misunderstanding of our Approach and the Literature (1/2)**
> > > > > >
> > > > > > We also still think that the reviewer has not understood the whole point behind the paper. We will try and explain again in the hope that the reviewer can understand why and face the issues in the understanding.
> > > > > >
> > > > > > ***Claims the method receives unstructured inputs as images***
> > > > > >
> > > > > > We would like to ask the reviewer to point out where we **claim** this. From line 39-42 we clearly state that **Given an input state, NUDGE extracts entities and their relations, converting raw states to a logic representation.** Also unstructured is used in our paper only in line 1. Again converting the raw inputs to logic representation is a classical technique done by neuro-symbolic systems so we do not understand the fixation on unstructured inputs here. According to the reviewer then, the paper **Mao et al, THE NEURO-SYMBOLIC CONCEPT LEARNER: INTERPRETING SCENES, WORDS, AND SENTENCES FROM NATURAL SUPERVISION, ICLR 2019** is also not neuro-symbolic since they also convert raw scenes to embeddings *with symbolic representations* using pre-trained  Mask-RCNN. This is just 1 paper, and there are 100s of others that convert raw scenes to a more abstract representation.
> > > > > >
> > > > > >
> > > > > > We provide a differentiable mapping from raw input to symbolic representations in a latent space. Of course, we provide the language for the latent space.
> > > > > >
> > > > > > ***this makes the problem simpler***
> > > > > >
> > > > > > We have already replied in detail about this in the 1st rebuttal. We use a logical representation obtained from the raw images as input.  Just because this does not adhere to the reviewer's definition of **neuro-symbolic and thus complex** does not make the statement true as we have already mentioned and shown in our 2nd reply.
> > > > > >
> > > > > > ***experimental setting is changed***
> > > > > >
> > > > > > It actually is not changed. We claim that NUDGE extracts entities and relations from raw inputs and our experiments follows the same.
> > > > > >
> > > > > > ***the method is not interesting***
> > > > > >
> > > > > > This is a personal opinion and the authors and other reviewers disagree with this.
> > > > > >
> > > > > > ***$\partial$ILP is unknown and Scallop exists***
> > > > > >
> > > > > > Well, just because the reviewer is not aware of a work, does not make it unknown.
> > > > > >
> > > > > > The most relevant work, Neural Logic Reinforcement Learning (NLRL), builds upon the $\partial$ILP framework. Their main idea is to compose policy functions using $\partial$ILP. We are surprised that the reviewer calls $\partial$ILP unknown and then giving a confidence of 4 with low score. We are actually speechless at this claim.
> > > > > >
> > > > > > Moreover, **$\partial$ILP is a structure-learning framework, i.e. they learn explicit rules as NUDGE does. Scallop is a parameter-learning framework, i.e. they learn continuous parameters (e.g. in NNs) given logical rules.** Thus Scallop is not applicable in our setting because we aim to learn explicit rules.
> > > > > >
> > > > > > Just for reference, **$\partial$ILP  won The Annual IJCAI-JAIR Best Paper Prize 2021 and has 481 citations**, and Scallop has 29 citations. This makes your argument null and void anyhow. Also, we know Scallop exists and have worked with it extensively, but in this paper there seemed no reason to include it.
> > > > > >
> > > > > >
> > > > > > We do not understand why you are opposing our work, because if you want to use Scallop or some of its follow-ups [see e.g. arxiv.org/abs/2306.08397]  in RL (no one has done that yet), then you require the NUDGE pipeline and making Scallop &  Co differentiable. We are in touch with the authors of these papers and know for the fact that there are ongoing efforts to make it happen.
> > > > > >
> > > > > > ***However, for new environments, I still think it is simple (compared with other environments used with structured inputs).***
> > > > > >
> > > > > > In our response to Reviewer AnR8, we have shown that NUDGE learns programs that are as complex as those learned by PIRL, another well-established RL agent that learns symbolic programs. We have shown and compared explicitly the obtained programs from both frameworks.
> > > > > >
> > > > > > Moreover, the Karen environment mentioned in the reviewer’s 1st answer does not involve any logical/relational constraints, and thus unclear how complex it is compared to our environments in terms of *logical reasoning*, which is the main focus of the paper.
> > > > > >
> > > > > > We believe that a clarified review, more than a feeling or thought with no sufficient specification, is necessary to set up and conduct additional experiments to improve the manuscript. Namely, we would like to see which exact environment provides the non-simpler setting to what degree, what is the gap from our current environment,  and how critical it is to be solved for the improvement of the manuscript. Otherwise, it is very challenging for us to integrate the reviewer’s feedback into the manuscript properly.

---

> > > > > > > ### Author Response · Authors · 2023-08-15
> > > > > > > **Misunderstanding of our Approach and the Literature (2/2)**
> > > > > > >
> > > > > > > ***"lame neural policy -> good neural guidance" and "actor critics v.s. policy gradients"***
> > > > > > >
> > > > > > > We would like the reviewer to explain how the provided answers are only intuitions and guesses?
> > > > > > >
> > > > > > > The 1st answer: **Neural policies are used to find the set of promising rules that NUDGE agents will use when learning to solve the task. Of course, the better the neural agents are, the more likely is the set of rule to be successful. NUDGE agents then have the advantage of relying on logic rules, and their embedded priors, which allows for faster and better generalization.** This is actually fundamental and also backed up by our experimental results.
> > > > > > >
> > > > > > > The 2nd answer: **We directly incorporated the actor-critic framework because we use the critic in the neurally guided symbolic abstraction. The critic stabilizes the gradients backpropagated to the policy compared to direct backpropagation from the return. The focus of our paper is not to compare the advantages of existing RL methods.** Again this is fundamental in RL as to why actor-critic approaches are preferred. These are not guesses. Also, *can the reviewer give reasons why comparing to policy gradient will enhance the results or make the paper better?*
> > > > > > >
> > > > > > > Thus the main issues in the review are: (1) hand-wavy claims in the review not backed up by solid evidence or reasons (2) fixation on unstructured inputs and wrongly claiming that our work misguides the reader. As we have shown, we make it explicitly clear what NUDGE does in the introduction. **We defer this to the AC and other reviewers to make a fair and unbiased call.**

---

> > > > > > > > ### Comment · Reviewer_AhLx · 2023-08-15
> > > > > > > >
> > > > > > > > I agree to defer the discussion to AC and other reviewers. I have tried my best to state my concerns clearly while being, hopefully, still polite.
> > > > > > > >
> > > > > > > > I do feel obligated to defend other great works that were mentioned in the authors' last reply, just for clarification and for futural reference.
> > > > > > > > * NSCL talks a lot about symbol grounding. It is undoubtedly neuro-symbolic. They do not assume semantics of objects/predicates to claim improved explainability, etc.
> > > > > > > > * $\partial$ILP is surely good work. I did not say it is unknown. I even mentioned it in one of my replies. There is nowhere in the authors' first rebuttal talking about $\partial$ILP, but I guess the authors used it as "classic" in their rebuttal, which can potentially explain why the authors thought I mentioned $\partial$ILP as unknown. In any case, both $\partial$ILP and scallop support gradient updates of their free parameters. There is no need to translate an RL problem to a supervised learning setting to compare with them.
> > > > > > > > * Scallop can of course support learning structures. Just for the authors' information, I have no relationship with scallop. I just mentioned it as "evidence" saying that there are more recent works and $\partial$ILP can be considered as old/classic.
> > > > > > > >
> > > > > > > > I will ignore many other personal insults to me and will let the others decide. I think I have done my job as a reviewer, trying to explain from so many perspectives in detail about the issues I am concerned with.

---

> > > > > > > > > ### Author Response · Authors · 2023-08-16
> > > > > > > > >
> > > > > > > > > We would like to apologize if the reviewer felt that there was a personal attack. We surely did not intend to do that and were pointing out factual errors in the review in the hope that the reviewer sees our point. We will do the same in this new reply (although as we all agree the AC and other reviewers are the best judges at this point).
> > > > > > > > >
> > > > > > > > > ***I guess the authors used $\partial$ILP as "classic" in their rebuttal***
> > > > > > > > >
> > > > > > > > > **No.**  We kindly point the reviewer to our first rebuttal. We have written:
> > > > > > > > >
> > > > > > > > > To demonstrate this, we conducted reasoning experiments using trained NUDGE agents and **Classic logic benchmark, where the set of action rules are given but with all weights set to 1.0, which simulates discrete logic.**
> > > > > > > > >
> > > > > > > > > We prepared this baseline because of the reviewer’s feedback: ***classic ILP methods may even work***. Classic ILP refers to Inductive Logic Programming frameworks using classic (discrete) logic in general, including FOIL, Aleph, Progol, and ILASP, to name a few. See:
> > > > > > > > >
> > > > > > > > > Andrew Cropper and Sebastijan Dumančić: Inductive Logic Programming At 30: A New Introduction, Journal of Artificial Intelligence Research, Vol. 74, 2022.
> > > > > > > > >
> > > > > > > > > To use these classic ILP systems, we need to transform RL tasks to binary classification tasks *because an ILP problem is defined as a tuple of (1) a set of positive examples, (2) a set of negative examples, and (3) a background knowledge, where each element is represented as logical atoms.*
> > > > > > > > >
> > > > > > > > >
> > > > > > > > > $\partial$ILP is NOT a classic ILP framework since it is a differentiable structure learner using gradients. And our classic baseline is not $\partial$ILP.
> > > > > > > > >
> > > > > > > > >
> > > > > > > > > ***Scallop can of course support learning structures.***
> > > > > > > > >
> > > > > > > > > **This is again a crucial factual error.** Scallop, in its current form, cannot learn explicit programs from data as NUDGE does. **Scallop optimizes perception neural networks given input logic programs i.e., they require logic programs as its input. We aim to learn explicit logic programs from data instead of perception networks.**
> > > > > > > > >
> > > > > > > > > These different learning schemes are called *parameter learning* and *structure learning*, respectively, in the ILP/neuro-symbolic community. In general, structure learning is a harder task since it involves discrete structure optimization. *The fact that a model can perform parameter learning does not entail that it can perform structure learning. In principle although Scallop can do structure learning (any logical framework can) but this is yet to be implemented and is a separate paper in itself.*
> > > > > > > > >
> > > > > > > > > We hope this clarifies the misunderstanding.
> > > > > > > > >
> > > > > > > > >
> > > > > > > > > ***NSCL talks a lot about symbol grounding. It is undoubtedly neuro-symbolic. They do not assume semantics of objects/predicates***
> > > > > > > > >
> > > > > > > > > We fully agree with the reviewer that NS-CL is a neuro-symbolic approach. However, NS-CL uses pre-defined operations (predicates) in their DSL, i.e., gives semantics of the programs as inductive bias. (for reference, Tab.9 in the NSCL paper's appendix shows exactly how each operation is implemented and given.)
> > > > > > > > >
> > > > > > > > > Thus we claim that using pre-defined language or semantics does not make NUDGE non-neuro-symbolic or a simple method in that regard. This is a counter-argument to the claim by the reviewer: *This paper claims it is and compares itself with Neuro-Symbolic methods. BUT it actually uses a Symbolic policy setting*.

---

> > > > > > > > > > ### Comment · Reviewer_AhLx · 2023-08-16
> > > > > > > > > >
> > > > > > > > > > Again, please do not diminish other papers to make this one look better. I will try to clarify some confusion about them from my understanding, hopefully for the last time:
> > > > > > > > > > * Regarding scallop, if it can learn a mapping from operator images to symbolic operators (i.e., the perception module), it can learn symbolic operators (thus programs) as free parameters as well.
> > > > > > > > > > * Regarding NSCL, they talk a lot about symbol grounding (i.e., binding semantics with objects/predicates through learning, Figure 3 & 4 of that paper). Jointly learning this binding with the logic structures is already much more difficult than learning just one. I cannot find a similar module for learning binding in this paper. I do hope I am wrong since that may well end this endless discussion.
> > > > > > > > > >
> > > > > > > > > > I will not repeat my other concerns.

---

> > > > > > > > > > > ### Author Response · Authors · 2023-08-16
> > > > > > > > > > >
> > > > > > > > > > > Again, please do not deviate from our answers. We for sure did not diminish any papers to make our paper look good. We are proud of our paper and this reviewer cannot take that away from us as just a personal opinion doesn't determine the quality of our work. The reviewer is the one that diminished $\partial$ILP in the first response, so this statement seems  not well founded. Of course blaming is easy but backing up claims with scientific evidence is difficult that's why this discussion is more about feelings at this point which is a bit sad.
> > > > > > > > > > >
> > > > > > > > > > > Also we agree that there is no point of discussion since the reviewer seems to think that only their way of thinking is correct. Of course your argument about Scallop holds no ground but it's clear there is no point in discussing this further.

---

### Official Review · Reviewer_NLZb · 2023-07-06

**Soundness:** 3 good
**Presentation:** 3 good
**Contribution:** 2 fair
**Rating:** 6
**Confidence:** 4

**Summary:**

This paper introduces NUDGE, a framework for interpretable and explainable policy reasoning and learning in reinforcement learning. NUDGE employs differentiable forward reasoning during inference to derive a set of weighted rules that form an interpretable policy. During training, NUDGE leverages neurally-guided symbolic abstraction to efficiently distill rule-based symbolic representations from neural-based policies. It then employs gradient-based policy optimization using actor-critic methods to learn the weights of these rules. Empirical results demonstrate that NUDGE achieves comparable performance to neural-based policies while providing interpretable and explainable logical representations. Furthermore, the rule-based symbolic policies in NUDGE exhibit adaptability and flexibility in addressing environmental changes. Overall, NUDGE offers a promising approach to achieving interpretable and adaptive policies in reinforcement learning.

**Strengths:**

+ NUDGE introduces end-to-end reasoning architectures based on differentiable forward reasoning, allowing for comprehensive policy generation directly from raw input.
+ Its neurally guided symbolic abstraction leverages a pre-trained neural policy to induce a set of candidate action rules, which are subsequently optimized using the critic component of an actor-critic agent.
+ Notably, NUDGE policies are differentiable, enabling the utilization of action gradients with respect to input atoms. This unique feature facilitates the explanation of state features that influence the decision-making process, shedding light on how specific actions are chosen in a given state.

These aspects collectively contribute to NUDGE's interpretability and explainability, providing insights into the underlying mechanisms of the learned policies.

**Weaknesses:**

It appears that the individual components of NUDGE are not inherently novel on their own. The paper's main contribution lies in the integration of existing components to form the NUDGE framework. By bringing together existing components such as differentiable forward reasoning, neurally guided symbolic abstraction, and policy optimization, NUDGE creates a framework for interpretable and explainable policy learning. Although the individual components are not novel on their own, the paper would have been acceptable if it can still demonstrate the novel insights that arise from their integration. However, the novelty of NUDGE in comparison to prior work such as PIRL [Verma et al., 2018] appears to be very limited. Both NUDGE and PIRL employ a pretrained neural policy as a guide for learning policies. Additionally, both NUDGE and PIRL utilize random search to update policy structures and employ optimization techniques to learn policy parameters. The primary distinction lies in the fact that NUDGE learns programmatic policies within logical programs, whereas PIRL learns policies within functional programs. While NUDGE can learn from raw image inputs, this aspect is not considered a core contribution, as it relies on off-the-shelf tools for object extraction from visual inputs.

**Questions:**

There is confusion regarding the paper's claim on policy explainability in relation to NUDGE. The approach used for explainability appears to be fairly standard and not inherently tied to differentiable logical programs. In fact, it can be applied to explain a neural network as well e.g. annotating regions of an image that mostly influence policy output. Therefore, it remains unclear how NUDGE specifically enhances policy explainability compared to existing methods. Could the authors elaborate on that?



**Limitations:**

Although the paper is well executed and integrates existing components effectively, it does not present advancements that significantly differentiate it from prior work PIRL [Verma et al., 2018].

---

> ### Author Rebuttal · Authors · 2023-08-09
>
> We thank the reviewer for the detailed feedback and for finding that the combination of differentiable reasoning and RL contributes to the interpretability and explainability issues in RL and provides insights into the underlying mechanism of RL agents.
>
> We address the concerns next.
>
> ***NUDGE limited novelty wrt PIRL.***
>
> The main difference is that PIRL develops programs based on restricted functional language, but NUDGE searches over a more general language of first order logic.
> PIRL requires *sketch* as input to synthesize programs, which is a specification of the structures of programs to be generated, then optimize the internal parameters in the sketch. In contrast, NUDGE performs structure learning from scratch using neural guidance on symbolic abstraction and weight optimization. We provide a language bias (mode declaration) to restrict the search space, but not the structure of the programs.
> As NUDGE uses first-order logic, it can incorporate background knowledge in a declarative form (e.g., with a few lines of relational atoms and rules), which is difficult with the functional programming language.
>
>
> PIRL nicely incorporates recursive calls of learned programs. We have not developed this for NUDGE, but labeling together combinations of logic atoms would ease the understanding of the rules. For example, creating an eatable atom in case of a 3fishes environment where fishes are only eatable if they are both green and smaller.
> However, NUDGE makes it easy to incorporate background knowledge (e.g. closeby)
>
> Finally, PIRL is developed for and evaluated on continuous/control tasks, e.g. TORCS, where the main interest is not on relational reasoning.
>
>
>
> ***How NUDGE specifically enhances policy explainability compared to existing methods.***
>
> NUDGE uses gradient-based explanation methods, which can compute explanations very efficiently using automatic differentiation.  Explainability is thus produced efficiently in comparison to classical logic-based policies (that are not differentiable and thus require additional hard coding to produce explanations).
>
> Moreover, heatmaps used on DNNs can highlight raw inputs and not over concepts. The symbolic abstraction of our logic policies allows us to differentiate between the importance of the size and of the color of a fish for the decision of, e.g., dodging. Heatmaps of DNNs would not be able to make such distinctions.
> Thank you for raising this point, it helped us improve the quality of our explanation on this topic.

---

> > ### Comment · Reviewer_NLZb · 2023-08-15
> > **Thanks for your response**
> >
> > I thank the authors for their thorough response. I kindly ask them to revisit the PIRL paper, particularly Section 3.1. In essence, PIRL optimizes both the program's structure and its internal parameters. Consequently, I perceive NUDGE's novelty to be somewhat limited, as it appears to be another iteration of the PIRL framework (though with distinctions in programming languages, i.e., functional programs vs. logical programs). I believe this limitation should be recognized in the paper's final version.

---

> > > ### Author Response · Authors · 2023-08-15
> > > **Thanks for the comment**
> > >
> > > We would like to thank the reviewer for the response. As asked by the reviewer we did go through the section 3.1 of the PIRL paper again and as the reviewer correctly notes and as we also mentioned in our1st reply PIRL does optimize over the specification of the structures of programs to be generated but the overall structure of these programs are fixed. This is different from the structure learning of NUDGE so even though the papers go in same direction and we agree with reviewer here, calling NUDGE "just" an iteration of PIRL will be a little unfair.
> > >
> > > We will anyhow add a discussion on this in the final version of the paper. Thank you for engaging with us. Let us know if there are any further concerns and we hope you can reconsider your rating.

---

> > > > ### Comment · Reviewer_NLZb · 2023-08-17
> > > > **Thanks for the explanation**
> > > >
> > > > I will keep my score as it is due to the following reasons:
> > > >
> > > > (1) In the field of program synthesis, we refer to the specification of program structures as a context-free grammar. This applies to any synthesis work, including your own, which utilizes context-free grammar to constrain the program search space. From this perspective, I don't consider NUDGE as significantly divergent from PIRL in terms of structure learning.
> > > >
> > > > (2) To me, a more effective approach could be to frame NUDGE as an instantiation of the PIRL framework. This would involve highlighting the difficulties of directly adapting the search algorithm in PIRL for learning logical programs and showcasing how this paper successfully addresses these challenges. Please note that this is my personal perspective, and the authors should feel free to convey their own ideas.

---

### Official Review · Reviewer_AnR8 · 2023-07-07

**Soundness:** 2 fair
**Presentation:** 3 good
**Contribution:** 2 fair
**Rating:** 5
**Confidence:** 2

**Summary:**

In this work, the authors propose neurally guided differentiable logic policies (NUDGE), which learns differentiable logical policies that are represented by interpretable rules. It is capable of producing explanations for their decisions in complex environments. NUDGE utilizes neurally guided symbolic abstraction to boost the model's performance. Specifically, the candidate rules for the logic-based agents are obtained efficiently by being guided by neural-based agents. Experimental results show that NUDGE can achieve competitive performance compared with the neural baselines. Besides, the NUDGE agent can adapt to environmental changes and its learned policies are interpretable and explainable.

**Strengths:**

1. The authors propose using a novel mechanism called neurally guided symbolic abstraction, to boost the neural logic model's performance.
2. The writing of the paper is well-organized and clear, and the evaluations are quite comprehensive.

**Weaknesses:**

It is unclear how are the abstract head atoms being interpreted, for example, **right_go_to_door** used in the GetOut task and **right_to_eat** in the Fishes tasks. Are they automatically learned or interpreted by developers manually?

**Questions:**

1. What's the input representation for the neural baselines, namely, DQN agents and PPO agents? Is it the same with the neural logic methods?
2. Wonder how does NUDGE perform under more logically challenging environments, for example, grid environments which are utilized in previous work [1][2].
3. How are the head atoms interpreted, for example, **right_go_to_door** used in the GetOut task and **right_to_eat** in the Fishes tasks. Are they automatically learned or interpreted by developers manually?

[1] Cao, Yushi, et al. "GALOIS: boosting deep reinforcement learning via generalizable logic synthesis." Advances in Neural Information Processing Systems 35 (2022): 19930-19943.
[2] Yichen Yang, Jeevana Priya Inala, Osbert Bastani, Yewen Pu, Armando Solar-Lezama, and Martin Rinard.
Program synthesis guided reinforcement learning for partially observed environments. Advances in Neural
Information Processing Systems, 34, 2021.

**Limitations:**

The authors have adequately addressed the limitations.

---

> ### Author Rebuttal · Authors · 2023-08-09
>
> We thank the reviewer for the thoughtful comment and for acknowledging that the paper proposes a novel neuro-symbolic RL method, and the writing and presentation are well-organized and quite comprehensive.
>
> We address the concerns next.
>
> ***Unclear how are the abstract head atoms being interpreted***
>
> As stated on lines 240-242, we renamed the predicates for clarity using the obtained explanatory rules after training.
> However, rules with interpretable atoms are automatically learned, and the renaming process does not require much additional resources and effort.
>
> Your point led us to think that these interpretations could be automatically identified. Let us illustrate how to do this.
> For example, after training, NUDGE obtained rules about the action to go right:
> * right$^{(1)}$(agent):-type(O1,agent),type(O2,door),on-right(O2,O1),has_key(O1).
> * right$^{(2)}$(agent):-type(O1,agent),type(O2,key),on_right(O2,O1),not_has_key(O1).
>
> As these two rules have the same semantic head atom, a program could automatically search for differences in the bodies of the rules. Here, the right$^{(1)}$ head could be relabelled right$^{(door)}$ and right$^{(2)}$ relabelled  right$^{(key)}$, as these rules are respectively concerned with objects of types door and key. We will add the discussion in the next version.
>
>
> ***Input representation for the neural baselines***
>
>
> Yes, we used object-centric representations for both neural and logic agents. Continuous values (e.g. x, y) are unchanged. For the logic agent, the categorical values are one-hot encoded to fit the αILP framework.
>
>
> ***NUDGE in more logically challenging environments***
>
>
> NUDGE can handle as complex rule as those used in the GALOIS paper [1]. For example,
> the synthesized programs shown in the paper [1] are:
> * (1.00) gt_box() :- ¬ has_key(X), is_agent(X), ¬ has_key(Y), is_env(Y)
> * (1.00) gt_key() :- ¬ has_key(X), is_agent(X), has_key(Y), is_env(Y)
> * (1.00) gt_door() :- has_key(X), is_agent(X), ¬ is_open(Y), is_door(Y)
> * (1.00) gt_goal() :- has_key(X), is_agent(X), is_open(Y), is_door(Y)
>
> These rules have similar complexity as the ones learned by NUDGE in our experiments  (e.g. loot)  in terms of the number of relations and variables. Thus, NUDGE will perform well in the environment used in GALOIS.
>
> To further reduce the gap between our environments and those from [1], we have adapted our loot environment (Loot-hard). In this environment, the agent first has to pick up keys and open their corresponding saves, before being able to exit the level (by going to a final exit tile).
> We provide the results hereafter, as well as curves on **Fig.1 in the attached PDF**.
> |   PPO   |  NUDGE  | Random  |
> |---------|---------|---------|
> | 8.4±3.2 | 9.2±2.0 | 1.2±0.4 |
>
>
> [1] GALOIS: Boosting Deep Reinforcement Learning via Generalizable Logic Synthesis. NeurIPS 2022

---

> > ### Author Response · Authors · 2023-08-16
> > **Looking Forward to Feedback on Our Response**
> >
> > Dear reviewer,
> >
> > We appreciate the time and effort that you have taken to provide us with the review. We would like to ask if the reviewer has any further concerns or is satisfied by our responses to the original review.
> >
> > We are looking forward to any further discussion with the reviewer and would like to thank the reviewer again for helping make our paper better.
> >
> > Regards,
> >
> > The Authors

---

> > > ### Comment · Senior_Area_Chairs · 2023-08-18
> > > **Please engage with rebuttal**
> > >
> > > The authors have posted their rebuttal—has it addressed your questions and concerns? Even if you don't find it convincing, it'd be helpful to at least acknowledge that you've read it. There's a pretty wide range of scores on this one, so I'd like to make sure we're being thorough.
> > >
> > > Thanks!

---

> > > ### Comment · Reviewer_AnR8 · 2023-08-19
> > >
> > > Thank you to the authors for their time and response.
> > >
> > > Overall, I appreciate the idea of using trained neural network-based agents to guide the search of logic rules. However, I agree that the authors should add more discussion about PIRL in their paper.

---

> > > > ### Author Response · Authors · 2023-08-19
> > > > **Thank you for the revised rating**
> > > >
> > > > We would like to thank the reviewer for appreciating the idea of our paper and raising the score. As already mentioned in the response to reviewer NLZb, we will add a discussion regarding PIRL in the final version of the manuscript.
> > > >
> > > > Thank you again for the  thoughtful comments and the raise in the score.

---

### Author Rebuttal · Authors · 2023-08-09

Dear reviewers,

Thank you all for your helpful feedback that helped us improve our manuscript's clarity and quality.


Here we provide a PDF that contains 2 figures:
* Figure 1: Returns (avg.±std.) obtained by NUDGE and neural PPO on the newly created more logically challenging environment (*Loot-hard*).
* Figure 2: Additional examples of explanations and action distributions produced by NUDGE for 2 different states of GetOut.

---

### Decision · Program_Chairs · 2023-09-21

**Decision:**

Accept (poster)

**Comment:**

This paper proses NUDGE: Neurally gUided Differentiable loGic policiEs. NUDGE requires a trained actor-critic model and a trained perception module, and uses those to guide the search for (interpretable) rules.

First an acknowledgement. The discussion of this paper was *extremely* contentious, and serves, in my opinion, as an example of how *not* to discuss a paper. I did my best to make my decision objectively in spite of this.

Most reviewers praised the proposed method for symbolic abstraction (and e.g. that the complexity and perhaps interpretability of the policies can be controlled), the writing and the evaluations (even though the experimental settings vary). Multiple reviewers found it unclear from the current draft that the method requires a *given* perception module, and given this the authors need to make that limitation very clear in the final manuscript.

While some reviewers point out that the components of NUDGE are not particularly novel on their own, the combination in the proposed framework still adds to, and deviates enough from, previous work such as PIRL in my opinion. (The authors should, as they promised in discussion, include more discussion regarding PIRL, but I am confident that they can do that.)

Despite much back and forth, the reviewers did not reach a consensus, yet 4/5 reviewers recommend acceptance to varying degrees, and I don't see enough reason to override that.